# Sono-activable and biocatalytic 3D-printed scaffolds for intelligently sequential therapies in osteosarcoma eradication and defect regeneration

Xiao Rong [1,7], Sutong Xiao [1,2,7], Wei Geng[2], Bihui Zhu[1], Ping Mou[3], Zichuan Ding[1], Boqing Zhang[4] ✉, Yujiang Fan[4], Li Qiu [1] ✉ & Chong Cheng [2,5,6] ✉

To mitigate the necessity for multiple invasive procedures in treating malignant osteosarcoma, an innovative therapeutic approach is imperative to achieve controllable tumor-killing effects and subsequent bone repair. Here, we propose the de novo design of sono-activable and biocatalytic nanoparticles-modified 3D-printed hydroxyapatite (HA) scaffold (HS-ICTO) for intelligently sequential therapies in osteosarcoma eradication and bone defect regeneration. The engineered HS-ICTO scaffold displays superior, spatio-temporally controllable $H_2O_2$-catalytic performances, which promptly generate massive reactive oxygen species via multienzyme-like mechanisms coupled with sono-activation, thus augmenting tumor cell apoptosis. Furthermore, HS-ICTO can intelligently switch to catalyze $H_2O_2$ to $O_2$ within the inflammatory bone defect microenvironment, effectively blocking endogenous $H_2O_2$-mediated oxidative stress, which positively modulates the osteogenic differentiation of stem cells and ultimately facilitates defect regeneration. We validate that this multifaceted HS-ICTO scaffold possesses robust and on-demand abilities to prevent neoplastic recurrence and promote anti-inflammatory osseous tissue repair, representing a promising platform for precision oncological intervention and regenerative medicine.

Osteosarcoma is a highly malignant bone tumor that mostly occurs in children and adolescents and mainly invades long bones. The aggressive behaviors of osteosarcoma can cause bone destruction, severe pain, disability, and even metastasis, which call for systematic medical and surgical interventions[1,2]. Limb salvage therapy, including complete tumor excision, bone defect reconstruction, and adjuvant chemo or radiotherapy, is commonly employed, which ideally can both remove osteosarcoma and restore limb function[3,4]. However, the presence of chemo-radio-resistance of osteosarcoma frequently impedes the attainment of microscopically margin-negative (R0) resections[5–8],

[1]Department of Medical Ultrasound, West China Hospital, Sichuan University, Chengdu, China. [2]College of Polymer Science and Engineering, National Key Laboratory of Advanced Polymer Materials, Sichuan University, Chengdu, China. [3]Orthopaedic Research Institute, Department of Orthopaedics, West China Hospital, Sichuan University, Chengdu, China. [4]College of Biomedical Engineering, National Engineering Research Center for Biomaterials, Sichuan University, Chengdu, China. [5]State Key Laboratory of Oral Diseases, National Center for Stomatology, National Clinical Research Center for Oral Diseases, West China Hospital of Stomatology, Sichuan University, Chengdu, China. [6]Institute of Chemistry and Biochemistry, Free University of Berlin, Berlin, Germany. [7]These authors contributed equally: Xiao Rong, Sutong Xiao. ✉e-mail: boqing_zhang@scu.edu.cn; qiulihx@scu.edu.cn; chong.cheng@scu.edu.cn

which is associated with unfavorable prognoses present as local recurrence and limb salvage failure following endoprosthetic reconstruction that may need a secondary or multiple surgical interventions[9–11]. This dilemma underscores the desperate need to develop an alternative therapy capable of delivering intelligent and controllable tumor-killing effects and sequentially promoting bone regeneration, thereby mitigating the necessity for multiple invasive procedures and ultimately enhancing patient prognosis[12].

Recent advancements in developing implantable and biomimetic 3D-printed composite scaffolds offer a promising strategy for limb salvage therapy[8,13–15]. By integrating 3D printing and nanotechnology, composite scaffolds can provide physical and biological support in the defect area for adhesion, proliferation, and osteogenic differentiation of the stem cells. Among them, calcium phosphate (CaP) ceramics scaffolds, including hydroxyapatite (HA) and beta-tricalcium phosphate (β-TCP), have emerged as a prominent substitute for bone grafts due to their striking similarity to the mineral composition of natural bone, along with their excellent biocompatibility, osteoconductivity, and osteoinductive properties that can ideally restore the morphology and function of the defect bone[16–18]. However, osteosarcoma invasion[19], coupled with hyperactive osteolysis triggered by tumor metabolism[20], and surgical resection, may cause extensive bone defects in an inappropriate regenerative microenvironment, which exceeds the intrinsic healing capabilities of such scaffolds. Current modification strategies employing bioactive factors[21–24] or stem cells[25,26] integrated into scaffolds overcome these limitations to some extent but also encounter insurmountable challenges, such as integrated tumor-killing and bone regeneration, intelligent microenvironmental adaptability, complex material design, and insufficient regenerative efficacy[27–29]. Considering the magnitude of these problems in limb salvage therapy, essential efforts should be devoted to designing straightforward and versatile composite scaffolds that are capable of intelligently adapting to the complex redox dynamics in both tumor-killing and bone regeneration microenvironments[30].

The tumor recurrence of osteosarcoma or inflammatory trauma after limb salvage therapy can lead to a significantly increased level of $H_2O_2$ and hypoxic microenvironments[31–36], which may severely disrupt the redox homeostasis and cause sustained oxidative stress to the endogenous stem cells[37,38]. Therefore, engineering specific materials that can simultaneously target $H_2O_2$-based biocatalysis for controllably elevating oxidative stress in osteosarcoma or modulating redox balance in inflammatory trauma represents an essential strategic goal. Ideally, the composite scaffolds should be able to intelligently transform $H_2O_2$ into highly toxic reactive oxygen species (ROS) to induce oxidative stress-mediated tumor cell death and decompose $H_2O_2$ to produce $O_2$ for alleviating the inflammatory responses and also restoring the osteogenesis-osteoclastogenesis balance[39]. Recent advances in nanotechnology have driven the rapid developments of enzyme-mimicking biocatalytic materials, which demonstrate significant potential in targeting $H_2O_2$-based biocatalysis for therapeutic applications[40–42]. However, these currently developed biocatalytic materials, such as metal oxides[43] and noble metal nanoparticles[44,45], are hard to fully adapt to the complex biochemical requirements of intelligently transforming $H_2O_2$ into ROS and $O_2$. Moreover, the established therapeutic strategies focus either on antitumor (generating ROS) or anti-inflammation (scavenging ROS); no biocatalytic materials have been designed for limb salvage therapy. Therefore, rationally developing comprehensive and versatile biocatalytic materials to eradicate tumor cells intelligently and remodel redox homeostasis becomes indispensable for limb salvage therapy.

Here, to overcome the enormous challenge of limb salvage therapy for osteosarcoma, we propose the de novo design of sono-activable and biocatalytic 3D-printed scaffolds with paradoxical but synergistic functions to achieve intelligently sequential therapies in osteosarcoma eradication and defect regeneration after surgical implantation (Fig. 1a). The intelligently therapeutic systems were composed of two objectives: (1) using $TiO_2$ nanomaterials to serve as semiconducting substrates to design Ir clusters with sono-activable and multienzyme-like properties via potent electronic coupling by Ti-O-Ir structure (named ICTO) for targeted $H_2O_2$ biocatalysis[46–48]; (2) efficient deposition of ICTO biocatalytic materials onto 3D-printed HA scaffold (named, HS-ICTO) to provide an integrated biocatalytic properties for osteosarcoma eradication, anti-inflammation, and mechanical support to synergistically promote bone defect repair. Notably, our studies reveal that under tumor microenvironment (TME) conditions, both the fabricated ICTO nanoparticles and HS-ICTO scaffolds demonstrate superior, spatiotemporally controllable, and versatile $H_2O_2$-catalytic performances. ICTO and HS-ICTO can promptly generate a substantial quantity of ROS in TME via both peroxidase (POD)- and oxidase (OXD)-like pathways in conjunction with sono-activation effects; meanwhile, they can also accelerate the catabolism of intracellular glutathione (GSH) and synergistically disrupt cellular redox homeostasis, which eventually induces mitochondrial damage and tumor cell apoptosis. Additionally, the catalase (CAT)-like activity facilitates the generation of $O_2$ to alleviate the hypoxic microenvironment and promote the sono-activable effects on tumor cell death (Fig. 1b). Furthermore, for the subsequent bone defect regeneration, HS-ICTO can intelligently switch to catalyze $H_2O_2$ to $O_2$ for effectively blocking endogenous $H_2O_2$-mediated oxidative stress and supply abundant $O_2$, which positively modulates the osteogenic differentiation of stem cells, suppresses the osteoclastogenesis process, and ultimately facilitates defect regeneration (Fig. 1c). Our research introduces a straightforward, controllable, and intelligently sequential therapeutic system for osteosarcoma by tackling the paradoxical concerns of tumor recurrence and bone regeneration with superior efficiency.

## Results

### Synthesis and characterization of HS-ICTO

A straightforward wet evaporation and deposition strategy was employed to integrate versatile ICTO nanoparticles onto 3D-printed HS to fabricate sono-activable and biocatalytic scaffolds (HS-ICTO) (Fig. 2a, Supplementary Figs. 1, 2a). Field-emission scanning electron microscopy (FESEM) was initially utilized to detect the surface morphology of HS-ICTO-x (where x = 0, 0.5, 1.0, 2.0 mg/mL of ICTO nanoparticles), revealing the meticulously interlocked and uniformly interwoven structures of HS (Supplementary Fig. 2b–d). Further magnified SEM images emphasized the surface deposition and coating of abundant ICTO on HS without substantially affecting the original scaffold architecture (Fig. 2b–d and Supplementary Fig. 2e). Moreover, the energy-dispersive X-ray spectroscopy (EDS) mapping discloses the homogenous existence of Ti, Ir, Ca, P, and O elements across the scaffolds, indicating the successful coating of ICTO (Supplementary Fig. 2f). To ascertain whether the mechanical properties of the HS-ICTO scaffold meet the standards for bone tissue implants, we employed a universal mechanical testing system. The findings reveal that HS and HS-ICTO demonstrate nearly identical deformation profiles. In particular, the variance in compressive strength between HS (2.01 ± 0.29 MPa) and HS-ICTO (2.02 ± 0.15 MPa) display no statistically significant difference, which suggests that the ICTO coating does not compromise the structural integrity or compressive properties of the HS matrix and is comparable to the mechanical properties of human trabecular bone (Supplementary Fig. 3). Notably, the petaloid morphology and hierarchical porous structure of ICTO, in conjunction with the porous structures and rough surface characteristics of 3D-printed HS, facilitate the robust interfacial adhesion and anchoring of ICTO. This interaction ensures the relatively good stability of ICTO nanoparticles on the HS substrate, which can withstand scouring, washing, and long-term soaking. Nanoparticle detachment is observed

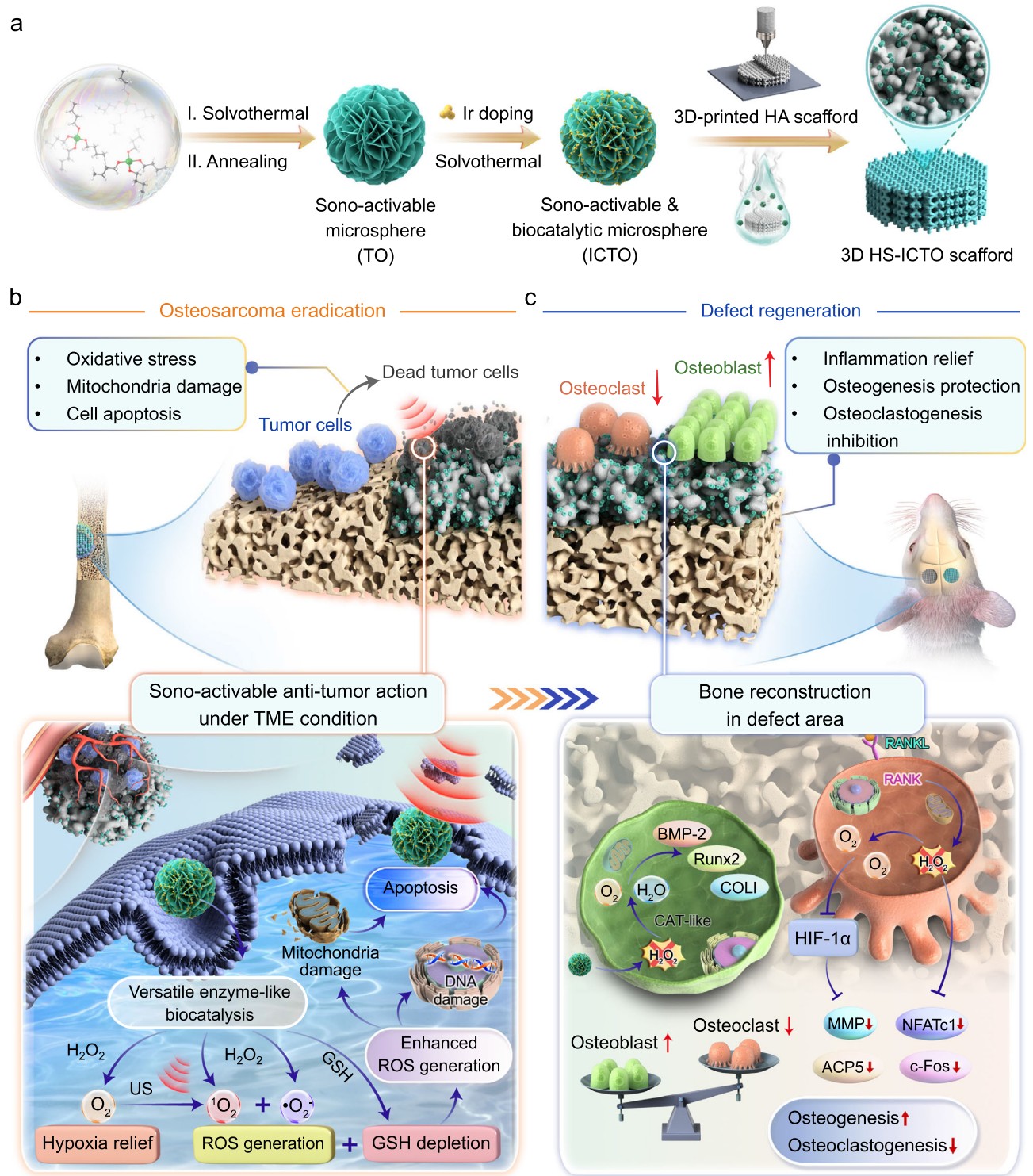

**Fig. 1 | Engineering sono-activable and biocatalytic 3D-printed scaffolds for limb salvage therapy of osteosarcoma. a** Synthesis of the ICTO and ICTO-modified 3D-printed HA scaffolds (HS-ICTO). **b** Sono-activable and versatile biocatalytic ROS generation in intelligent and controllable osteosarcoma eradication. TME indicates tumor microenvironment. **c** Excess endogenous $H_2O_2$ scavenging and redox homeostasis remodeling from the defect area for superior bone reconstruction. BMP-2 bone morphogenetic protein-2, Runx2 Runt-related transcription factor-2, COLI type I collagen, RANKL receptor activator of nuclear factor κB ligand, HIF-1α hypoxia-inducible factor 1-alpha, MMP matrix metalloproteinase, NFATc1 nuclear factor of activated T-cells, cytoplasmic 1, ACP5 acid phosphatase 5.

exclusively under mechanical scraping or external energy disturbance (Supplementary Fig. 4).

Given the essential function of ICTO as the active component within the composite scaffold, we proceeded to elucidate its detailed microstructural and compositional characteristics. X-ray diffraction (XRD) pattern of ICTO exhibits anatase structure conforming to the standard pattern of pristine $TiO_2$ (named TO)[49,50], with no diffraction peaks of Ir phase, which signifies ultralow Ir incorporation (approximately 2.6 wt%) (Supplementary Fig. 5)[51,52]. Subsequent SEM and transmission electron microscopy (TEM) images display that both pristine TO and ICTO possess a characteristic multi-edged petaloid microstructure and are uniformly distributed (Fig. 2e and

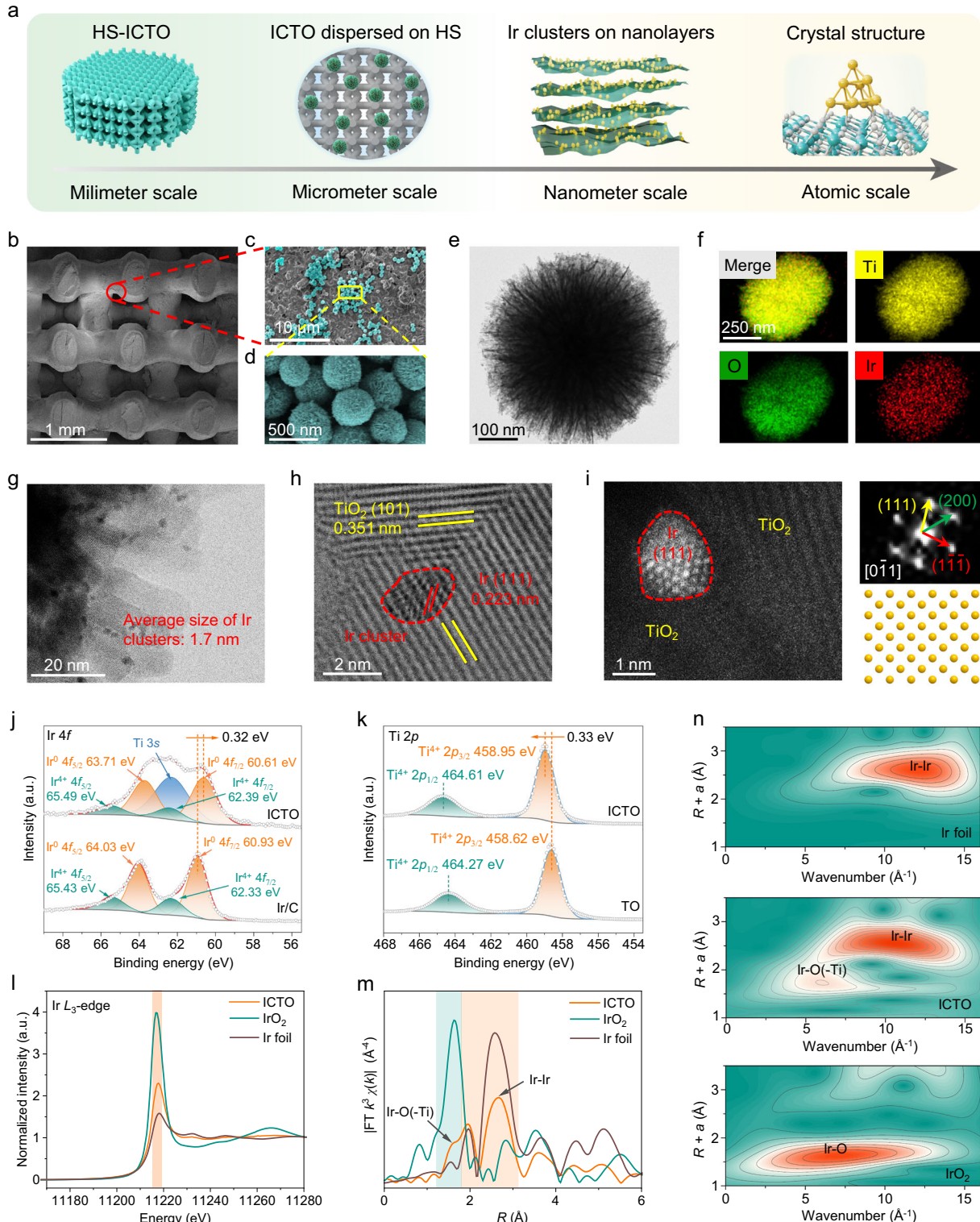

**Fig. 2 | Morphology and structural characterization. a** Structure illustration of HS-ICTO scaffold at different scales from millimeter to atomic scale (color codes: gray, O; cyan, Ti; yellow, Ir). **b** Representative SEM images and **c**, **d** magnified SEM images of HS-ICTO. **e** TEM images of ICTO. **f** The corresponding EDS elemental mapping of ICTO. **g** HRTEM images of ICTO. **h** Magnified HRTEM images of ICTO. **i** HAADF-STEM image of ICTO with the corresponding FFT pattern. **j** XPS spectra of ICTO and Ir/C in Ir 4f regions. **k** XPS spectra of ICTO and TO in Ti 2p regions. **l** Normalized XANES spectra at Ir $L_3$-edge. **m** FT $k^3$-weighted FT spectra in (k)-

function of the EXAFS spectra for the Ir L-edge. **n** WT images for the $k^3$-weighted EXAFS signals. R represents the distance between the adsorbing atoms and neighboring atoms, and χ(k) denotes the amplitude of the EXAFS oscillations as a function of photoelectron wavenumber k. The color gradient delineates the transition from high signal intensity (orange) to low signal intensity (cyan). In (j–l), a.u. indicates the arbitrary units. Experiments were repeated independently (b–i) three times with similar results. Source data are provided as a Source Data file.

Supplementary Fig. 6). Concurrently, the corresponding EDS mapping corroborates the uniform distribution of Ir species (Fig. 2f). Meanwhile, an amplified version of the edge nanosheets unveil the formation of highly dispersed Ir clusters with an average size of approximately 1.7 nm on TO supports (Fig. 2g and Supplementary Fig. 7). The high-resolution TEM (HRTEM) images further deliver detailed lattice stripes with interplanar spacings of 0.223 and 0.351 nm, well-matched to (111) planes of Ir and (101) plane of anatase $TiO_2$ nanocrystals in ICTO, respectively (Fig. 2h and Supplementary Fig. 8)[53]. Afterward, high-angle annular dark-field scanning TEM (HAADF-STEM) was performed to validate the atomic-scale distribution of Ir species, illustrating abundant well-distributed Ir clusters across TO support (Supplementary Fig. 9). The atomic arrangements and interplanar spacing of Ir clusters are visually investigated by atomic-resolution HAADF-STEM images, with the associated fast-Fourier transform (FFT) pattern validating the structure of Ir cluster (Fig. 2i and Supplementary Fig. 10)[54].

Thereafter, X-ray photoelectron spectroscopy (XPS) measurement was performed to probe the electronic state and coordination environment of the fabricated ICTO. The XPS survey spectra distinctly affirm the predominance of Ti, O, and Ir in ICTO (Supplementary Fig. 11). The high-resolution Ir $4f$ spectrum of ICTO demonstrates two doublets centered at 60.61 eV/63.71 eV and 62.39 eV/65.49 eV, ascribed to metallic $Ir^0$ and oxidized $Ir^{4+}$, respectively[55–57], which implies the formation of Ir-O linkages at the edges of Ir clusters and TO support (Fig. 2j). Notably, an obvious negative shift of metallic Ir species of 0.32 eV compared with carbon-supported Ir sites (Ir/C) are revealed. Accordingly, the Ti species in ICTO display a positive valence compared with pristine TO (Fig. 2k), thus suggesting the existence of robust Ir-O-Ti interactions and the occurrence of electron transfer between the Ir clusters and $TiO_2$, which may affect its biocatalytic performances.

We then carried out X-ray absorption spectroscopy (XAS) analysis to investigate the local bonding microstructure of ICTO. X-ray absorption near edge structure (XANES) spectra at Ir $L$-edge display higher white lines (WL) absorption intensity than that of Ir foil and significantly lower than that of $IrO_2$, suggesting an average valence between $Ir^0$ and $Ir^{4+}$ (Fig. 2l and Supplementary Fig. 12). Besides, Fourier-transformed (FT) and wavelet-transforms (WT) of $k^3$-weighted extended X-ray absorption fine structure (EXAFS) spectra at Ir $L_3$-edge exhibit a predominant Ir-Ir coordination (~2.5 Å) and weak Ir-O-Ti coordination (~1.75) (Fig. 2m, n), which primarily demonstrate the formation of interphase structures between Ir clusters and TO substrate[58]. To sum up, the strong Ir-O-Ti chemical coupling and potent interfacial charge transfer between Ir clusters and TO supports may promote the stabilization and catalytic redox properties of ICTO materials.

## Enzyme-like biocatalysis evaluation and mechanism analysis

Leveraging the distinctive 3D hierarchical petaloid nanosheet microstructure with optimized specific surface area, ICTO guarantees superior $H_2O_2$ substrate accessibility, optimal exposure of surface Ir active sites and enhanced ultrasonic energy absorption, which may result in favorable reactivities in sono-activable ROS biocatalytic processes (Fig. 3a). The POD- and OXD-like activities were primarily evaluated utilizing the universal 3,3′,5,5′-tetramethylbenzidine (TMB) colorimetric assay[59–61]. The ICTO showcases superior POD- and OXD-like activities in comparison with pristine TO (Fig. 3b and Supplementary Fig. 13). Moreover, the pH-dependent POD- and OXD-like activities have also been detected to demonstrate that ICTO only presents significant ROS-generated activity in mildly acidic conditions (similar to the pH values in TME), thus selectively responding to the TME without inflaming normal tissues (Supplementary Figs. 14, 15). Subsequently, steady-state catalytic kinetics was comprehensively explored, including the catalytic constant ($K_m$), the maximal reaction

velocity ($V_{max}$), and the turnover number (TON)[62–64]. According to the calculated results, compared to pristine TO ($V_{max} = 0.18 \times 10^{-7}$ M s$^{-1}$ and TON $= 2.77 \times 10^{-5}$ s$^{-1}$), the ICTO exhibits higher $V_{max} = 3.51 \times 10^{-7}$ M s$^{-1}$ and TON $= 21.60 \times 10^{-3}$ s$^{-1}$, alongside a lower $K_m$ ($K_m = 0.655$ mM) (Fig. 3c, Supplementary Figs. 16–18, and Supplementary Table 1).

Additionally, ICTO exhibits superior enzymology indexes across many other earlier reported POD mimics, including metal nanoparticles, metal oxides, single-atom-based enzyme mimics, and organic frameworks-based enzyme mimics (Supplementary Fig. 19 and Supplementary Table 2). Besides, we also examined the glutathione (GSH) depletion capacity of ICTO, as illustrated in Fig. 3d and Supplementary Fig. 20. A significant decrease in the UV absorbance intensity of 5,5′-dithiobis-(2-nitrobenzoic acid) (DTNB) over time denotes the degradation of GSH to glutathione disulfide (GSSG), which suggests that ICTO can prevent the ROS quenching induced by GSH overexpression, thus amplifying the therapeutic effect[23]. Besides, the catalytic performance of ICTO for rapid $O_2$ production in the presence of $H_2O_2$ at pH 6.5 has been verified to simulate the TME condition (pH 6.5) of the tumor (Supplementary Fig. 21), which indicates that ICTO can not only alleviate the hypoxic microenvironment of TME but also supply $O_2$ to enhance ultrasonic irradiation-activated ROS production.

Following that, the generated ROS species by ICTO have been identified by free radical quenching experiments, which confirms that the major types of ROS are •$O_2^-$ and $^1O_2$ (Supplementary Fig. 22)[65]. Whereafter, we deployed 5,5-dimethyl-1-pyrroline N-oxide (DMPO) and 2,2,6,6-tetramethylpiperidine (TEMP) as the spin trap reagents (Fig. 3e), along with specific hydrazine (HE) radical probe (Supplementary Fig. 23), to visually elucidate the generation of •$O_2^-$ and $^1O_2$[66], as well as the enhancement effects of US irradiation. Additionally, the utilization of 9,10-diphenanthraquinone (DPA) indicator to detect $^1O_2$ demonstrates a marked reduction in the characteristic absorption peak with prolonged US exposure, implying a sustained $^1O_2$ generation facilitated by US (Supplementary Fig. 24). Furthermore, the noticeable increase in the oxidation level of TMB induced by US irradiation further visually emphasizes the augmenting effect of US irradiation on ROS generation (Supplementary Fig. 25).

Besides tumor eradication, bone defect regeneration is required, but great challenges are still faced during osteosarcoma treatment. It has been reported that the hypoxic microenvironment and the surplus of $H_2O_2$ resulting from inflammatory responses and metabolism in defective tissues can severely disrupt the bone reconstruction process[12,60]. Thus, effective regulation of $H_2O_2$ and $O_2$ levels in defective tissue with neutral pH is also essential to establish a more conducive microenvironment for bone healing, balancing oxidative stress and oxygen availability, and ultimately promoting optimal bone regeneration[35,67]. Consequently, we systematically explore the CAT-like ability of ICTO to decompose excess $H_2O_2$ and transform $H_2O_2$ to generate $O_2$ under neutral conditions[68,69]. Noticeably, ICTO shows an impressively increased $H_2O_2$-eliminating activity (about 90% within 20 min) compared to the negligible removal ability of pristine TO (9.37%). Alternatively, the dissolved $O_2$ concentration indicates a time-dependent pattern that correlated with the decomposition of $H_2O_2$ (Fig. 3f and Supplementary Fig. 26). Notably, ICTO exhibits superior kinetic parameters with a $V_{max}$ of 48.23 μM s$^{-1}$ and a TON of 4.11 s$^{-1}$ (Fig. 3g, Supplementary Fig. 27, and Supplementary Table 3). Furthermore, we evaluated the long-term activity and stability of ICTO, which reveal the consistent preservation of its morphology and sustained high-efficiency in $H_2O_2$ biocatalysis without appreciable deterioration (Supplementary Fig. 28). Taken together, we have demonstrated that the ICTO nanoparticles not only effectively and controllably produce massive ROS under TME conditions but also possess the capability to eliminate excess $H_2O_2$ and sustainably provide $O_2$ within the defective area after surgical treatments, which further highlights its exceptional potential for intelligently sequential therapies in osteosarcoma eradication and defect regeneration.

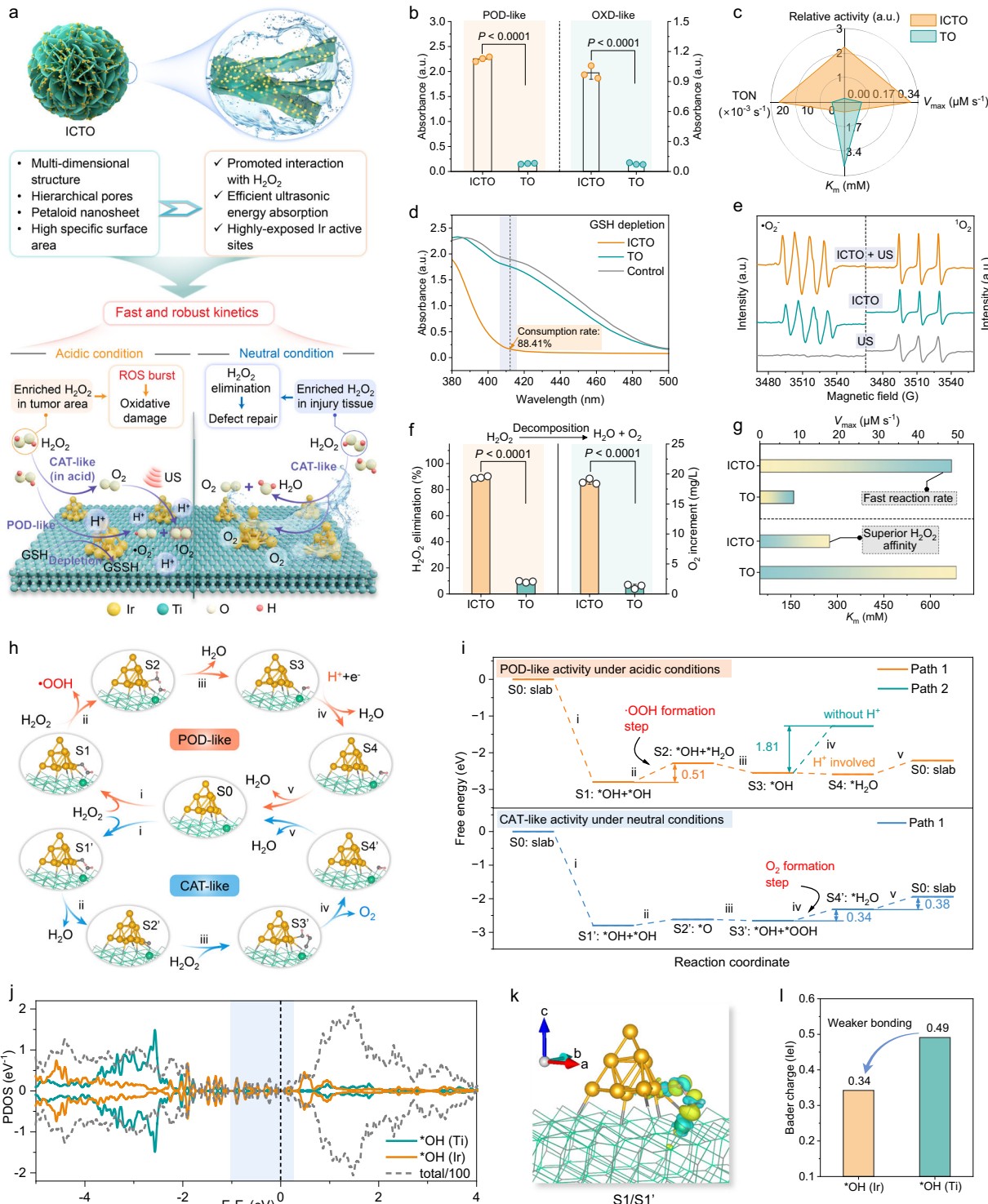

**Fig. 3 | Enzyme-like biocatalytic activities and theoretical calculations of ICTO.** **a** A comprehensive schematic representation of the advantageous structural characteristics of ICTO and the resultant versatile enzyme-like biocatalytic activities. **b** POD-like and OXD-like activities utilizing TMB assay at a wavelength of 652 nm after incubation with ICTO and TO ($n = 3$ independent experiments, data are presented as mean ± SD). **c** Comparison of kinetic parameters in ICTO and TO. **d** GSH depleting abilities with DTNB as the trapping agent of -SH in GSH. **e** The EPR spectra of the in situ $\cdot O_2^-$ radical detection by DMPO and $^1O_2$ radical detection by TEMP. **f** The $H_2O_2$ elimination and $O_2$ generation capacities of ICTO and TO under neutral conditions ($n = 3$ independent experiments, data are presented as mean ± SD). **g** A comparative analysis of the $V_{max}$ and $K_m$ parameters for ICTO and TO.

**h** Proposed reaction pathways and **i** corresponding Gibbs free energy diagram of POD-like and CAT-like catalytic pathways on ICTO (color codes: gray, O; green, Ti; orange, Ir; red, H). **j** Computed PDOS of O $2p$ orbital of *OH (Ir) and *OH (Ti) in ICTO. **k** Differential charge density analysis of ICTO*2OH (cyan and yellow are employed to denote charge depletion and accumulation, respectively; the cut-off of the density-difference isosurface is 0.01 $e \cdot Bohr^{-3}$). **l** Calculated Bader charge of *OH (Ti) and *OH (Ir) in ICTO. The assessment in (**b**, **f**) of $P$-values was performed by a two-tailed Student's $t$-test; all tests were two-sided. $V_{max}$ is the maximal reaction velocity, $K_m$ is the Michaelis constant, TON is the turnover number. In (**b**–**e**), a.u. indicates the arbitrary units. Source data are provided as a Source Data file.

To uncover the multifaced enzyme-like biocatalytic activities of ICTO, we executed comprehensive density functional theory (DFT) calculations to discover the generation processes of $\cdot O_2^-$ and $O_2$. Preliminary calculations using constructive modeling suggest that substantial electron transfers were mediated by Ir-O-Ti interfacial bonds between Ir clusters and TO supports (Supplementary Fig. 29). The proposed two reaction pathways reveal that following the initial adsorption of $H_2O_2$, it undergoes self-decomposition either on adjacent Ti and Ir atoms (pathway 1) or on two neighboring Ir atoms (pathway 2) as the reaction commences. As illustrated in Supplementary Figs. 30, 31, pathway 2 encounters significant energy barrier steps during subsequent POD-/CAT-like reactions, thus impeding the process. Consequently, we contend that the Ti sites within the TO supports are pivotal in optimizing the adsorption of oxygen intermediates. Analyzing the optimized reaction pathway demonstrates that during the POD-like ROS generation process, Ir*OH reacts with another $H_2O_2$ to release $\cdot OOH$ with an energy barrier of 0.51 eV, recognized as the rate-determining step (RDS). Followingly, $H_2O$ desorbs from the Ir site, retaining the initial *OH on the Ti site, which in turn captures an $H^+$ proton from the acidic environment to form $H_2O$ and depart from the surface, resetting the catalyst for the next cycle (Fig. 3h, i). Comparatively, in the absence of $H^+$ during the reaction, *OH interacts with another $H_2O_2$ to generate $*H_2O$ and *OOH intermediates. The subsequent desorption of *OOH encounters a significant thermodynamic energy barrier of 1.81 eV, emphasizing the necessity of an acidic environment in promoting the biocatalytic activity for optimal ROS production, which aligns well with prior experimental outcomes (Supplementary Figs. 32, 33). In addition to the POD-like pathway, we have also systematically carried out further calculations on the CAT-like pathway under neutral conditions. It is apparent that the desorption of $*O_2$ and $*H_2O$ from the catalytic sites serves as the RDS, with relatively small energy barriers of 0.34 eV and 0.38 eV[70-72], respectively, which suggests a superior CAT-like activity.

As mentioned, since Ir*OH and Ti*OH serve as pivotal intermediate states that appear at the initiation of POD-like and CAT-like reactions, we explored the intrinsic structure-reactivity relationship between their electronic structures and the multienzyme-like catalytic activities. Utilizing projected electron density of states (PDOS) to analyze the $p$-band of *OH (Fig. 3j), we find *OH (Ir) maintains elevated density states near the Fermi energy level relative to *OH (Ti), indicating greater activity and possibly a higher likelihood of subsequent reactions. Furthermore, comparisons of charge density differences (Fig. 3k and Supplementary Fig. 34) and Bader charges (Fig. 3l) disclose a higher charge transfer of 0.49 eV for *OH (Ti) in contrast to 0.34 eV for *OH (Ir). This implies a weaker Ir-O interaction than Ti-O, which further suggests that *OH located on the Ir site exhibits a higher potential to dissociate and react with another $H_2O_2$. Taken together, these findings underscore the essential function of Ir-O-Ti synergetic sites with reduced oxygen intermediate affinity in addressing the challenges posed by oxygen-involving multi-electron reactions inherent in the ROS-catalytic pathways.

## In vitro therapeutic effects in osteosarcoma eradication

First, the biocompatibility of bare ICTO nanoparticles on bone stem cells was carried out using the Cell-Counting-Kit-8 (CCK-8) assay at a predetermined concentration. As illustrated in Supplementary Fig. 35, both ICTO and TO nanoparticles demonstrate negligible cytotoxicity towards primary-cultured rat bone marrow mesenchymal stem cells (BMSCs, pH 7.4 medium) at concentrations up to 150 μg/mL, maintaining cellular viability above 95%. Conversely, exposure of 143b osteosarcoma cells to 150 μg/mL ICTO under simulated TME conditions (pH 6.5 medium) results in a significant reduction of cell viability to 45.3%, suggesting that ICTO can specifically respond to the acidic microenvironment of tumor cells, thereby inducing oxidative stress overload and apoptosis of tumor cells. The 2,7-dichlorofluorescein

diacetate (DCFH-DA) staining demonstrates a significant rise in the intracellular ROS level in the ICTO group. US irradiation (1 W/cm², 1.0 MHz, 30% duty cycle, 1 min) upregulates the ROS level in the TO group, and ICTO + US treatment presented a potent synergistic effect on increasing the ROS level (Supplementary Fig. 36). Moreover, the GSH-depletion effect of ICTO was confirmed through ThiolTracker Violet staining (Supplementary Fig. 37). Thus, we confirm that ICTO nanoparticles can promote ROS accumulation through a synergistic biocatalytic and sono-activation effect.

Next, the cellular therapeutic efficiencies of the biocatalytic and sono-activable 3D-printed scaffolds, HS-ICTO, were also investigated. Figure 4a illustrates the proposed therapeutic mechanism of HS-ICTO, which involves the conversion of $H_2O_2$ to $\cdot O_2^-$ (POD-like behavior) and $O_2$ (CAT-like behavior) under TME conditions, coupled with sono-activation effects to rapidly generate substantial ROS in the TME. Concurrent GSH catabolism synergistically disrupts cellular redox homeostasis. The observed multifaceted enzyme-like activities and ultrasound-amplifying effects of HS-ICTO align with those of bare ICTO (Fig. 4b, c and Supplementary Figs. 38–44). Besides, extended durability analyses demonstrate that HS-ICTO maintains continuous, efficient, and stable ROS generation capacity, further highlighting its considerable potential for prolonged therapeutic applications (Supplementary Fig. 45). The effects of ICTO loading amounts on HS scaffolds (HS-ICTO-0.5, 1.0, and 2.0) have also been studied, which present negligible ion-mediated cytotoxicity and no impairment against the BMSCs' viability (Supplementary Figs. 46, 47) while showing a dose-dependent manner on inhibiting 143b osteosarcoma cells. Notably, only 9.7% can be retained in the HS-ICTO-2.0 + US (1 W/cm², 1.0 MHz, 30% duty cycle, 1 min) group (Supplementary Fig. 48). Therefore, HS-ICTO-2.0 was chosen as the representative scaffold for the following detailed tumor-killing assessment.

Thereafter, the bulk RNA-sequence (RNA-seq) was used to investigate the different effects on gene expression after the HS-ICTO and HS-ICTO + US treatments. The principal component analysis (PCA), volcano plots, and Venn diagram all demonstrate the significant discrepancy of the gene expression profiles among the HS, HS-ICTO, and HS-ICTO + US groups (Fig. 4d, e and Supplementary Figs. 49, 50)[73]. The Gene Ontology (GO) term enrichment analysis reveals that the upregulated gene expression is predominantly associated with processes related to apoptosis, ROS damage, and stress response (Fig. 4f). The genes implicated in these GO terms are graphically represented in heat maps (Supplementary Fig. 51). Notably, the expression of genes involved in cholesterol and sterol metabolism process increases in HS-ICTO + US group compared with HS-ICTO alone. In this situation, the US irradiation combined with HS-ICTO acts as an unfavorable extrinsic clue that can activate the cholesterol biosynthesis in cancer cells for cellular adaptation and better survival[74].

Figure 4g demonstrates the confocal laser scanning microscopy (CLSM) observation and the corresponding 3D reconstruction images of live (green) and dead (red) 143b osteosarcoma cells, which were seeded on the surface of HS and HS-ICTO. The 143b cells can attach and distribute evenly on the surface of the HS scaffold within 24 h, and bare US irradiation did not significantly affect the live cell numbers (Supplementary Fig. 52). Following HS-ICTO treatment, a significant reduction in viable cell numbers is observed, accompanied by a concomitant increase in the number of dead cells. Upon combination with US irradiation, hardly any live cells could be found. Notably, an obvious decrease in total cell counts has been evident in both the HS-ICTO and HS-ICTO + US groups. This reduction may be attributed to cellular shrinkage during the apoptotic process, potentially facilitating detachment from the scaffold surface, thereby rendering them difficult to identify through CLSM observation. To further verify the apoptosis ratios of the 143b cells under different treatments, flow cytometry analysis was performed. As shown in Fig. 4h, the HS-ICTO group increases the apoptosis ratio of the Annexin-V positive

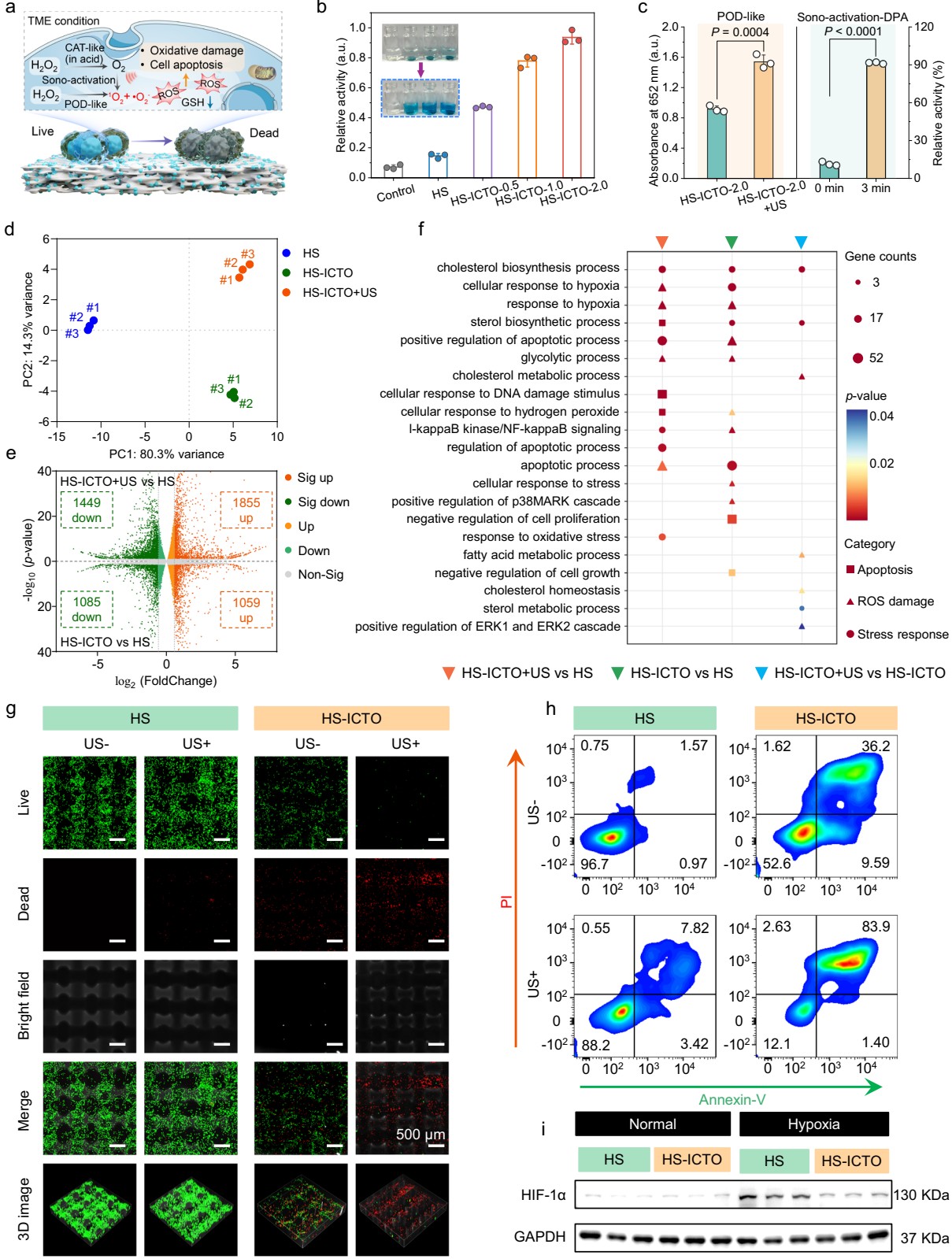

population from 4.43 to 45.08%, which is further elevated to 87.13% in combination with US irradiation (Supplementary Figs. 53, 54).

Hypoxia in TME can dramatically inhibit sono-activation efficacy because $O_2$ is the key reactant during the US treatment to generate $^1O_2$. The GO enrichment analysis presents that the gene expression related to the response to hypoxia is upregulated. However, the hypoxia-inducible factor 1-alpha (HIF-1α) expression, one of the most sensitive

genes to hypoxia, is downregulated in both HS-ICTO (fold change = 0.56, $P < 0.0001$) and HS-ICTO + US (fold change = 0.66, $P < 0.0001$) groups. Moreover, as shown in Fig. 4i and Supplementary Fig. 55, the hypoxia culture induction significantly improves the protein expression level of HIF-1α in the HS group while suppressing it in the HS-ICTO group. Thus, the upregulation in "response to hypoxia" GO terms may result from the ROS-triggered cellular damage that is independent of

**Fig. 4 | In vitro therapeutic effects in osteosarcoma eradication. a** Schematic illustration of the therapeutic mechanism of HS-ICTO under TME condition. **b** POD-like activity by evaluating the TMB absorbance value at λ = 652 nm after incubation with HS-ICTO-x (*n* = 3 independent experiments, data are presented as mean ± SD). **c** Sono-activable biocatalytic oxidation of TMB and DPA under the co-incubation of HS-ICTO (*n* = 3 independent experiments, data are presented as mean ± SD). **d** PCA plot for different cellular samples. **e** Volcano plots showing differential gene expression in HS-ICTO + US vs. HS and HS-ICTO vs. HS comparison (Sig up, significantly upregulated; Sig down, significantly downregulated; Up, upregulated; Down, downregulated; Non-Sig, nonsignificance). **f** GO enrichment analysis of the differentially expressed genes associated with ROS-triggered damage, stress response, and apoptosis biological process in 143b cells. The data in (**d**–**f**) were representative of three biologically independent samples from each group.

**g** Representative CLSM images of the live/dead staining of 143b cells on the scaffolds after different therapies. The images were representative of three independently repeated experiments from each group. **h** Flow cytometric apoptosis analysis of Annexin V-FITC/PI-stained 143b cells harvested from scaffolds with different treatments. The gate strategy is shown in Supplementary Fig. 53. The plots were representative of three independently repeated experiments from each group. **i** Expression of HIF-1α and glyceraldehyde-3-phosphate dehydrogenase (GAPDH) in 143b cells harvested from different scaffolds analyzed by western blot assay (*n* = 3 biologically independent replicates). The assessment in (**c**) of *P*-values was performed by a two-tailed Student's *t*-test. In (**e**), *P*-values were obtained from two-sided DESeq2 test without multiple comparison. In (**f**), *P*-values were obtained from one-sided Hypergeometric test without multiple comparisons. In (**b**, **c**), a.u. indicates the arbitrary units. Source data are provided as a Source Data file.

HIF-1α, which is also enriched in apoptosis and glycolysis GO terms[75,76]. Next, the DCFH-DA staining presents that the 143b cells on the HS-ICTO are positively stained, and the HS-ICTO + US stimulation demonstrates the highest fluorescence integrated density than other groups (Supplementary Fig. 56). Therefore, the HS-ICTO scaffold can relieve the hypoxia condition as well as support the sono-activated intracellular ROS generation effect.

The ROS-triggered intracellular damages were observed through TEM observation (Supplementary Fig. 57). It is found that the cells harvested from the HS and HS + US groups demonstrate a healthy and intact cellular and mitochondrial morphology. Meanwhile, ICTO nanoparticles are also found intracellularly, which indicates that the ICTO nanoparticles can undergo a fine cellular uptake process. Notably, the mitochondrial morphology presents profound swelling and partial cristae disarrangement in the HS-ICTO group. Moreover, the cells in the HS-ICTO + US group exhibit severe structural destruction along with mitochondrial cytolysis and bursting. Considering that accumulated ROS can trigger a decrease in mitochondrial membrane potential (MMP), we used the JC-1 as an indicator that forms aggregates with red fluorescence on normal MMP while residing as monomers with green fluorescence on abnormal MMP. According to Supplementary Fig. 58, the HS-ICTO + US group presents the lowest mean fluorescence intensity (MFI) of JC-1 aggregate and the highest MFI of JC-1 monomer compared to other groups. Overall, these results unequivocally confirm the multi-pathway enzyme-mimetic and sono-activated co-generation of ROS effects of HS-ICTO, leading to boosted intracellular ROS generation and promoting irreversible mitochondrial and cellular damage.

## In vivo therapeutic effects in osteosarcoma xenograft

Encouraged by the excellent in vitro therapeutic effects in osteosarcoma eradication, we further verified the synergistic sono-activation and biocatalytic effects of HS-ICTO on tumor-killing therapeutics using the 143b tumor-bearing mice. As displayed in Fig. 5a, the 143b osteosarcoma xenograft model was established in female Balb/c nude mice. Once the tumor volume reached ≈200 mm³, the HS and HS-ICTO were surgically implanted. Localized US irradiation (2.5 W/cm², 1 MHz, 30% duty cycle, 5 min) was applied at 24, 48, and 72 h after the implantation. The body weights and tumor volumes were recorded every third day during the therapeutic period. The possible in vivo antitumor mechanisms are presented in Fig. 5b, which states the synergistic therapeutic effects of HS-ICTO. The tumor growth is suppressed in the HS-ICTO and HS-ICTO + US groups compared to the sham group (Fig. 5c–f). The HS-ICTO + US group shows the most effective in tumor suppression, up to 90.43%. Of note, the HS + US somehow restrains the growth rate of the tumor xenograft. Considering the potential piezoelectric attributes of HA, we employed an oscilloscope and atomic force microscopy to investigate its piezoelectric properties (Supplementary Fig. 59). The results indicate a negligible presence of piezoelectricity in both HS and HS-ICTO. Thus, we hypothesize that the minor tumoricidal effects observed in HS and HS + US groups may be attributed to the subtle chemical effects of nano-HA, potentially involving calcium ion

overload[5,77]. The in vivo biocompatibility of HS and HS-ICTO was verified through weight measurement, organs hematoxylin and eosin (H&E) staining, and hemolytic test, which all support the healthy status of the mice after treatments (Fig. 5g, Supplementary Figs. 60, 61). Moreover, to further underscore the biosafety profile of HS-ICTO, we conducted subcutaneous implantation of HS and HS-ICTO in the dorsal region of 8-week-old male Sprague-Dawley (SD) rats. Peripheral blood was collected after 4 weeks for comprehensive hematological and biochemical evaluations of liver and kidney functions (Supplementary Fig. 62). Complete blood count analysis demonstrates that all parameters across both groups remain within the normal physiological range, with no significant inflammatory or anemic alterations observed. Furthermore, liver/kidney function biochemical assays reveal no notable abnormalities in critical indicators, thereby substantiating the circulatory safety of HS and HS-ICTO over extended metabolic periods.

The histological evaluations were performed to reveal the structural and cellular damage in tumor tissue (Fig. 5h–j). Similar H&E manifestations of the tumor tissue from the sham, HS, and HS + US groups are confirmed. In contrast, the HS-ICTO and HS-ICTO + US groups display typical histopathological damages, characterized by condensed nuclei, reduced cytoplasmic size, and tissue structure destruction. To verify the efficacy of HS-ICTO in alleviating tumor hypoxia, we performed HIF-1α immunofluorescence staining (Supplementary Fig. 63). The results demonstrate that, compared to the HS group, the expression levels of HIF-1α protein in tumor tissue within the HS-ICTO group are significantly reduced. Furthermore, the terminal deoxynucleotidyl transferase dUTP nick-end labeling (TUNEL) verifies the highest DNA damage ratio in the HS-ICTO + US group, indicating effective cellular damage. To further validate the apoptotic status of cells within the tumor tissue, we employed cleaved caspase-3 immunofluorescence staining, which specifically labels apoptotic cells (Supplementary Fig. 64). The results of the positive cell rate analysis reveal no significant differences in cleaved caspase-3-positive cells among the Sham, HS, and HS + US groups. In contrast, the HS-ICTO group exhibits a marked increase in the number of positive cells, and the HS-ICTO + US group shows nearly complete occupancy by positive cells. Combined with the TUNEL staining results, these findings robustly demonstrate that the HS-ICTO + US treatment effectively induces tumor cell apoptosis. Conversely, immunofluorescence staining for Ki67, a cell proliferation marker, demonstrates that the HS-ICTO + US treatment is the most effective at suppressing proliferation. Taken together, these in vivo assessments proved that the HS-ICTO effectively relieves the hypoxia condition of TME and facilitates biocatalytic and sono-activatable intratumoral ROS generation, which supports the excellent synergistic antitumor therapeutic effect of HS-ICTO + US irradiation.

## In vitro redox homeostasis modulation and stem cell protection

The surgical trauma (inflammatory responses and metabolism in defect) or tumor invasion of osteosarcoma can lead to a significantly increased level of H₂O₂ and hypoxic microenvironments, which may

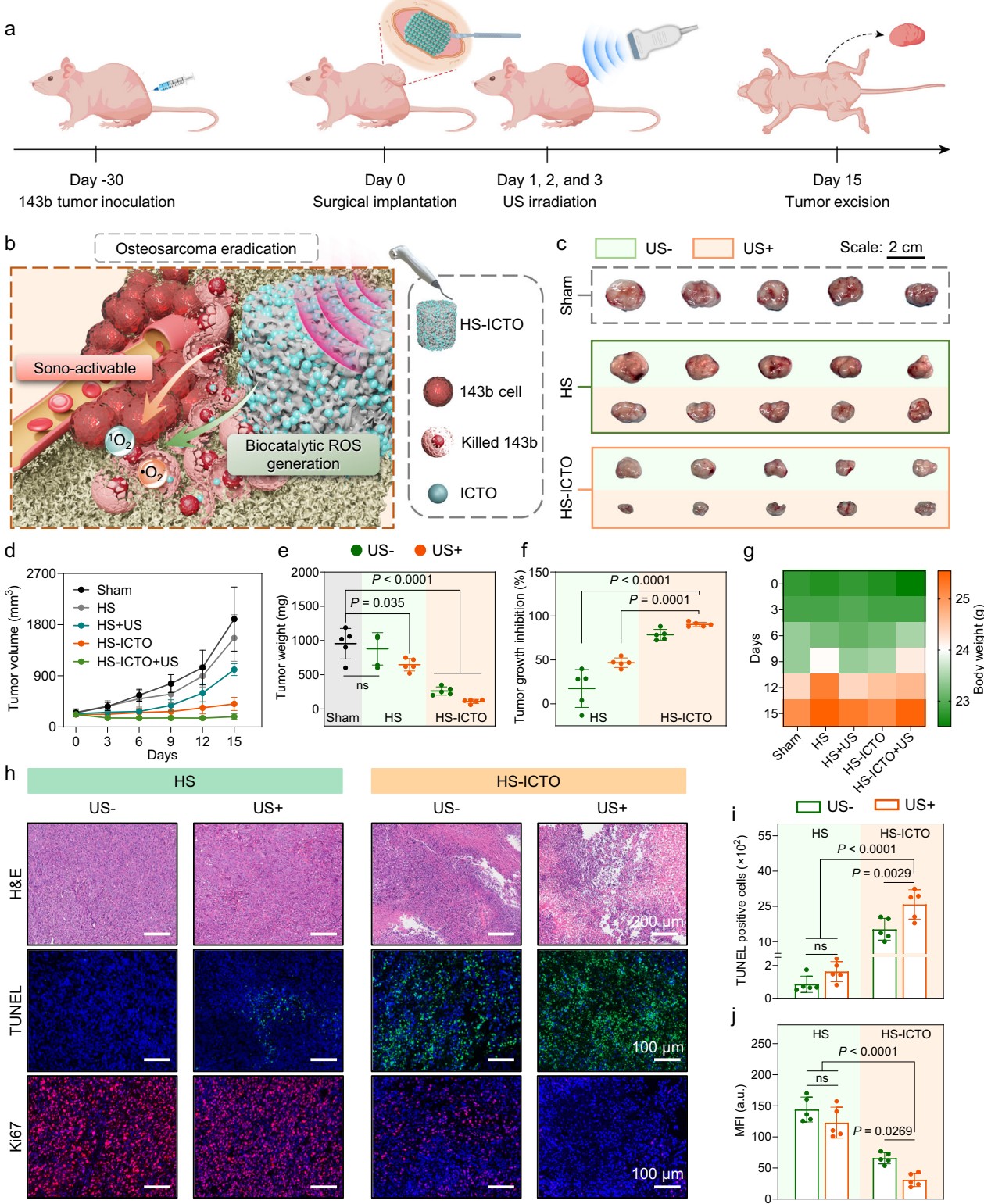

severely disrupt the redox homeostasis and cause sustained oxidative stress to the endogenous stem cells, thus dramatically impairing the bone regeneration process[30,78]. In general, the excessive $H_2O_2$ and insufficient $O_2$ supply may lead to cascaded side effects on the osteogenesis process of BMSCs by restraining motility, suppressing calcium deposition, and dampening the osteogenic expression[79,80]. Accordingly, since our HS-ICTO scaffold shows broad-spectrum CAT-

like activity in both acidic and neutral conditions, we then systematically assessed its CAT-like activities on $H_2O_2$-scavenging and $O_2$ evolution (Fig. 6a). As indicated in Fig. 6b, c, HS-ICTO has showcased excellent capacity to decompose $H_2O_2$ with elevated $O_2$ concentrations. This observation further underscores that the CAT-like activity depends on the loading amount of ICTO. Subsequent investigations into the temporal stability of HS-ICTO's $H_2O_2$ decomposition catalysis

**Fig. 5 | In vivo osteosarcoma xenograft killing effect. a** Illustration of 143b osteosarcoma xenograft establishment and the treatment procedure. **b** Schematic diagram of the osteosarcoma eradication mechanism of HS-ICTO. **c** Digital photos of the osteosarcoma xenograft after excision on day 15. **d** Tumor volumes are recorded every third day ($n = 5$ biologically independent mice per group; tumor volumes on day 15 were used for statistical analysis). **e** The tumor xenograft weights were measured on day 15 after excision ($n = 5$ biologically independent mice per group). **f** The tumor growth inhibition ratios ($n = 5$ biologically independent mice per group). **g** Mouse weight change presented as a heat map ($n = 5$ biologically independent mice per group). **h** The H&E, TUNEL, and Ki67 fluorescence staining images of the tumors from different groups. The fluorescence images were representative of three independently repeated experiments from each group. **i** Quantitative analysis of the TUNEL-positive stained cell numbers ($n = 5$ biologically independent replicates). **j** Quantitative analysis of the MFI of Ki67 staining ($n = 5$ biologically independent mice per group). Data are presented as mean ± SD, and ns represents no significant difference; statistical significance was calculated using one-way ANOVA followed by Tukey's post hoc test for multiple comparisons; all tests were two-sided. In (**j**), a.u. indicates the arbitrary units. Source data are provided as a Source Data file.

reveal consistent activity with negligible performance variation even after 4-week exposure to simulated physiological conditions, emphasizing its viability for sustained therapeutic interventions (Supplementary Fig. 65).

Thereafter, we conducted a meticulous assessment of HS-ICTO applied in guarding BMSCs against $H_2O_2$-induced oxidative stress. First, the proliferation and viability assay of BMSCs on HS-ICTO were performed. The BMSCs were seeded onto the scaffolds and exposed to either the control or $100\,\mu M$ $H_2O_2$-conditioned medium, and the adherent BMSCs were labeled by Calcein-AM staining on days 1, 3, and 5. As depicted in Fig. 6d and Supplementary Fig. 66, without $H_2O_2$, the BMSCs exhibit even spreading across the HS scaffold while undergoing contraction and reduction in cell number with $H_2O_2$ influence. Encouragingly, the live cell counts and viability of BMSCs on the HS-ICTO are not compromised by $H_2O_2$-mediated disruption. DCFH-DA staining confirms the intracellular ROS scavenging effect of HS-ICTO (Supplementary Fig. 67). To further verify the cellular spreading patterns on different scaffolds, we labeled the cytoskeletons with phalloidin-Rhodamine (red), followed by CLSM observation (Fig. 6e). Obviously, the cell spreading area in the HS group is significantly dampened by $H_2O_2$ that appeared to be rounder, slender, and exhibits fewer protrusions. However, the HS-ICTO maintains the spreading patterns of BMSCs under $H_2O_2$ conditions[81–83]. Additionally, the motility of BMSCs was analyzed through transwell-based migration assay (Supplementary Fig. 68). The results demonstrate that the migrated cells are reduced in the HS + $H_2O_2$ group, whereas the HS-ICTO + $H_2O_2$ group shows a significant improvement in the crystal violet-labeled migrated cells. This confirms that HS-ICTO can protect the migration ability of BMSCs under $H_2O_2$ stimulation. Next, the TEM observation confirmed that the uptake of ICTO nanoparticles from the HS-ICTO group would not interfere with the organellar and cellular structures of BMSCs (Fig. 6f). The BMSCs in the HS + $H_2O_2$ group present endoplasmic reticulum dilatation, mitochondrial swelling, and cristae disarrangement. In contrast, these morphological damages cannot be observed in the HS-ICTO + $H_2O_2$ group, suggesting a sufficient biocatalytic decomposition effect on $H_2O_2$.

Next, we carried out the osteogenic evaluation of BMSCs in different conditions. The early-stage osteoblast differentiation was detected by alkaline phosphatase (ALP) staining, and the late-stage mineralization was stained by Alizarin Red S (ARS). On day 14, the $H_2O_2$ stimulation significantly inhibits the ALP-positive area in the HS group, while HS-ICTO presents a rescuing effect (Fig. 6g, i). Similarly, on day 21, the $H_2O_2$ fails to impair the calcium deposition in the HS-ICTO group (Fig. 6h, j). Notably, the ALP expression and calcium deposition decrease significantly without scaffold co-incubation, which supports the osteogenic effect of the HA component (Supplementary Figs. 69, 70). Furthermore, immunofluorescence staining and RT-qPCR were performed to reveal the osteogenic expression and anabolism of BMSCs after different treatments. Runt-related transcription factor-2 (Runx2) is an essential transcription factor for osteogenic differentiation, and bone morphogenetic protein-2 (BMP-2) is another important bone formation inducer. The MFI of these markers is evidently decreased in the $H_2O_2$-treated HS group while relieved by HS-ICTO on day 14 (Fig. 6k, l). Notably, HS-ICTO can neutralize the negative effects of $H_2O_2$ on type I collagen (COLI) expression, which is the crucial

component of the extracellular matrix of bone tissue. Similar to immunofluorescence staining, the mRNA expression of Runx2, BMP-2, COLI, and ALP are rescued by HS-ICTO after $H_2O_2$ stimulation (Fig. 6m). Similarly, the group without scaffold induction demonstrates decreased expression of the above osteogenic markers (Supplementary Figs. 71, 72), which support the osteoinductive ability of HS. Taken together, we conclude that HS-ICTO can effectively protect the BMSCs from $H_2O_2$ and $H_2O_2$-related oxidative stress and preserve their cellular functions in bone defects, including viability, motility, and osteogenic ability (Fig. 6n).

## Suppression of osteoclastogenesis

The development of bone destruction and metastasis are severe complications associated with increased osteoclast activity[20]. In addition, this hyperfunction of osteoclasts can also impair the postoperative regeneration of bone defects. As the osteoclastogenesis involved $H_2O_2$ and HIF-1α-dependent signal pathway[84,85], we suppose the HS-ICTO may inhibit osteoclast differentiation via efficient $H_2O_2$ decomposition and $O_2$ supply, thus providing a "two birds with one stone" method for the signal pathway inhibition on ROS and hypoxia, which may eventually restrain bone resorption and benefit bone regeneration. First, we used a transwell system to verify the interaction between K7M2 cells (a murine osteosarcoma cell type) and RAW264.7 cells (a murine macrophage cell line) (Fig. 7a). After co-culturing for 7 days, the osteoclasts related expression of RAW264.7 cells are significantly increased, hence supporting the assertion that osteosarcoma cells can stimulate osteoclast differentiation (Fig. 7b). The possible anti-osteoclastogenesis mechanism of HS-ICTO is presented in Fig. 7c. Next, the RAW264.7 cells were stimulated by receptor activator of nuclear factor κB ligand (RANKL, a protein that stimulates the osteoclast differentiation of macrophage) and HS or HS-ICTO extracts. The phalloidine-FITC (green) staining confirms that in the positive control group (PC, only stimulated by 50 ng/mL RANKL), multiple RAW264.7 cells undergo fusion to form a giant cell, characterized by numerous nuclei (blue, labeled by DAPI) surrounded by an actin ring, which is the standard morphology of osteoclasts (Fig. 7d). The HS fails to suppress the formation of osteoclasts while in the HS-ICTO group the osteoclasts' size significantly decreases (Fig. 7e). Moreover, we performed the tartrate-resistant acid phosphatase (TRAP, an osteoclast-specific enzyme marker) staining after different treatments. As expected, compared with the PC and HS group, the TRAP-positive cells evidently decrease in the HS-ICTO group (Fig. 7f, g). The bare TO-loaded scaffold (HS-TO) extracts do not present significant impairment against the osteoclastogenesis process (Supplementary Fig. 73).

In the osteoclastogenesis process, the nuclear factor of activated T-cells, cytoplasmic 1 (NFATc1), is the master regulator, and c-Fos is the necessary transcription factor. Meanwhile, the matrix metalloproteinase-9 (MMP-9) and acid phosphatase 5 (ACP5) are crucial enzymes that help break down the bone matrix expressed by osteoclasts. The mRNA expressions of these crucial osteoclast indicators were analyzed through RT-qPCR. As presented in Fig. 7h, HS-ICTO suppresses the relative expression levels of the NFATc1 and c-Fos to 0.32 and 0.57, respectively, and the MMP-9 and ACP5 expressions are inhibited to 0.43 and 0.18, respectively. HIF-1α has

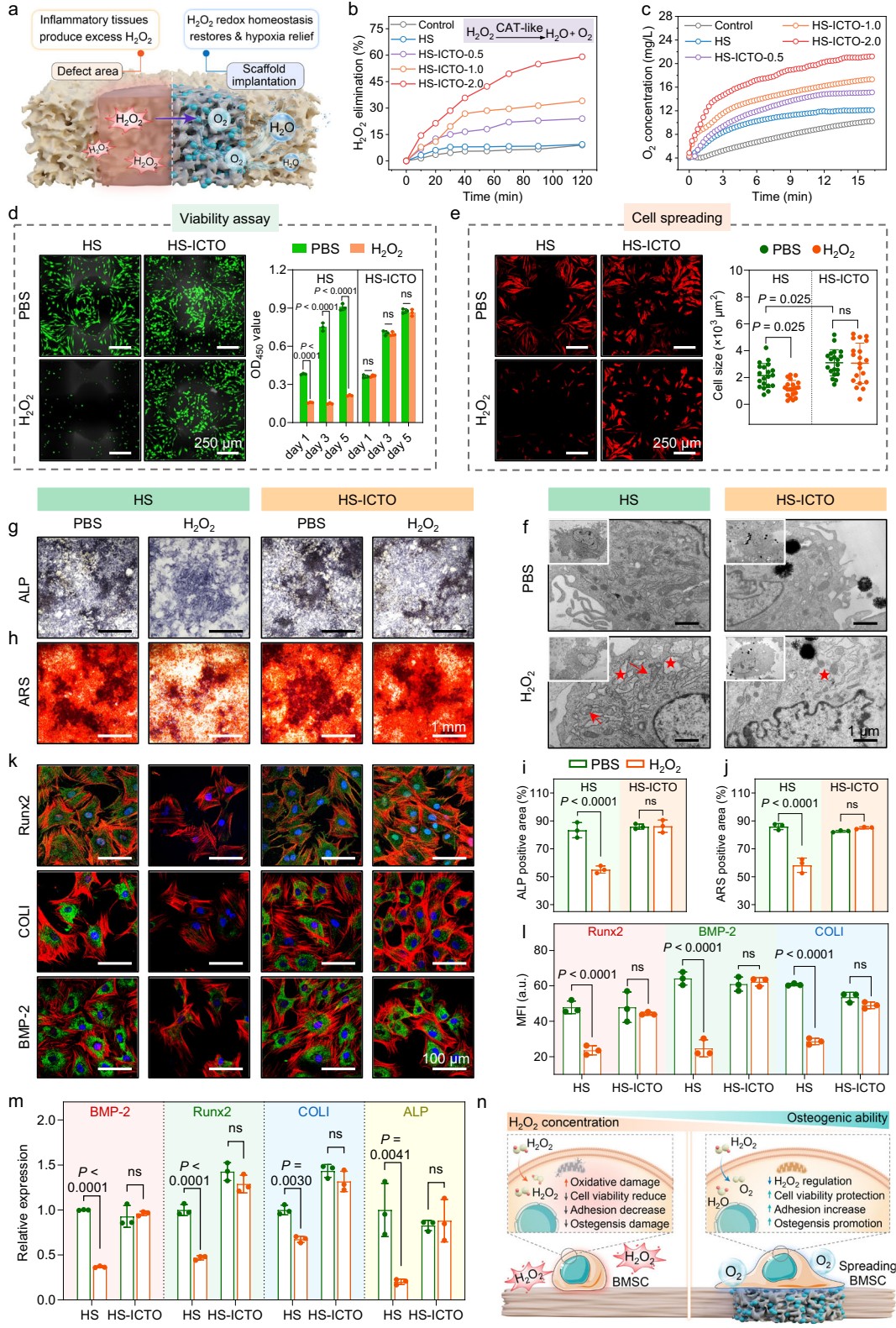

been reported to be required for osteoclast activation[84]. Here, in our western blot test, we find that the HIF-1α is significantly expressed by RANKL-stimulated RAW264.7 cells (Fig. 7i). However, only HS-ICTO treatment exhibits a suppression effect in HIF-1α expression. This phenomenon may result from the CAT-like activity of HS-ICTO to generate $O_2$ from the $H_2O_2$ produced by RANKL stimulation. Therefore, the HS-ICTO can effectively mitigate

osteosarcoma-associated bone destruction and benefit the repair of bone defects by suppressing osteoclast activity.

## Promotion of bone defect reconstruction

Motivated by the excellent osteogenic protection and osteoclasto-genesis inhibition effects of HS-ICTO 3D-printed scaffold, we then carried out the in vivo bone defect reconstruction assessments

**Fig. 6 | In vitro redox homeostasis modulation and stem cell protection against oxidative stress. a** Schematic illustration of the $H_2O_2$ redox homeostasis restore and $O_2$-induced hypoxia relief of HS-ICTO under bone defect condition. **b** Time-dependent $H_2O_2$ elimination of HS and HS-ICTO-x (0.5, 1.0, and 2.0). **c** The produced $O_2$ concentration was measured in the presence of HS and HS-ICTO-x (0.5, 1.0, and 2.0) and $H_2O_2$. **d** Representative CLSM images of live/dead staining of BMSCs seeded on 3D scaffolds with different treatments on day 5 and the corresponding CCK-8 assay result ($n = 3$ biologically independent replicates). **e** Representative CLSM images of phalloidin-Rhodamine stained BMSCs with different treatments and the corresponding cell size evaluation ($n = 20$, the size of twenty random cells from each group was calculated by ImageJ software). **f** TEM observation of the BMSCs after different treatments. The red arrow indicates damaged mitochondria, and the red star indicates swollen endoplasmic reticulum. **g** ALP staining of the BMSCs stimulated by osteogenic medium for 14 days with different treatments. **h** ARS staining of the BMSCs stimulated by osteogenic

medium for 21 days with different treatments. **i** Quantitative analysis of the ALP and **j** ARS positive staining area ($n = 3$ biologically independent replicates). **k** Immunofluorescence staining of Runx2, BMP-2, and COLI expression of BMSCs stimulated by osteogenic medium for 14 days with different treatments. The images in (**d–h**, **k**) were representative of three independently repeated experiments from each group. **l** Quantitative analysis of the MFI of Runx2, BMP-2, and COLI immunofluorescence staining ($n = 3$ biologically independent replicates). **m** The Runx2, BMP-2, COLI, and ALP gene expression of BMSCs after different stimulation for 14 days measured by RT-qPCR ($n = 3$ biologically independent replicates). **n** Schematic illustration of HS-ICTO regulating $H_2O_2$ redox homeostasis and protecting BMSCs. Data are presented as mean ± SD, and ns represents no significant difference; statistical significance was calculated using one-way ANOVA followed by Tukey's post hoc test for multiple comparisons; all tests were two-sided. In (**l**), a.u. indicates the arbitrary units. Source data are provided as a Source Data file.

through the rat cranial defect model. Briefly, twelve rats were anesthetized, and sagittal incisions on both sides of the scalp were produced. After that, the HS (left) and HS-ICTO (right) 3D-printed scaffolds were implanted into the defect area ($\varphi = 5$ mm). The repaired bone tissue was extracted for bulk RNA-seq at week 4. The radiographic and pathological assessments were performed at weeks 4, 8, and 12 (Fig. 8a).

PCA analysis and volcano plot demonstrate pronounced expression profile differences between the HS and HS-ICTO groups (Fig. 8b and Supplementary Fig. 74). Figure 8c shows the main downregulation GO enrichment terms and these differentially expressed genes are sorted out and presented in heat maps (Fig. 8d). It has been found that HS-ICTO treatment significantly suppresses the gene expression that related to inflammatory responses, including *CXCL6* and *CCL21* for neutrophil chemotaxis[86]; *TNF* and *IL1B* for positive regulation of I-kappaB kinase/NF-kappaB signaling[87]; bone resorption (*NFATC1* and *OSCAR* for osteoclast differentiation[88]; *MMP14* and *MMP9* for collagen catabolic process[89]); and ROS-related responses (*PRKAA1* and *GPX1* for response to hydrogen peroxide[90,91]; *ABCC1* and *BTG1* for response to oxidative stress[92,93]). It is noticeable that the genes related to "response to hypoxia" were also downregulated (*HIF1A*: fold change = 0.58, $P = 0.009$), which supports the hypoxia-relief function of HS-ICTO during bone regeneration.

The H&E and Masson's trichrome staining prove that at weeks 4 and 8, there is more immature collagen deposition (red arrows) and new bone formation (red stars) in the HS-ICTO group (Fig. 8e, f). In week 12, the HS-ICTO intra-scaffold area is filled by the newborn bone, but there is still immature collagen filling in the HS group. Intuitionally, the CLSM observation of Calcein-AM (green) and Alizarin red (red) labeled newborn osseous tissue at week 8 demonstrates a better osteoconductive property of HS-ICTO compared with HS (Supplementary Fig. 75). Besides, both scaffolds present great osteoinductivity when implanted subcutaneously (Supplementary Fig. 76). The 3D reconstructed micro-CT images demonstrate that HS-ICTO presents a relatively faster bone regeneration performance than HS at week 12 (Supplementary Fig. 77). Moreover, the coronal views present more osseous tissue ingrowth in the HS-ICTO scaffolds (Fig. 8g).

The bone regeneration-related parameters were evaluated using CTAn software. The bone volume (presented as bone volume versus tissue volume ratios (BV/TV)) has been promoted by HS-ICTO at week 8 (31.3 % vs. 23.2 %, $P = 0.0217$) and 12 (43.6 % vs. 28.7 %, $P = 0.0060$) (Fig. 8h). In the aspect of trabecular bone microarchitecture, the trabecular numbers (Tb.N) are significantly elevated by HS-ICTO treatment at week 12 (0.40 mm$^{-1}$ vs. 0.31 mm$^{-1}$ $P = 0.0084$) (Fig. 8i). Moreover, the bone mineral density (BMD) of the HS-ICTO group is higher compared with the HS group at each time point; and the trabecular separations (Tb.sp) are significantly suppressed at week 12 (Supplementary Fig. 78), signifying superior bone regeneration capacity of HS-ICTO. The intra-scaffold expressions of Runx2 (green) and

BMP-2 (red) are visualized by immunofluorescence staining, and COLI (gray) is labeled to present the spatial information. As presented in Fig. 8j–l, at each time point, the Runx2 and BMP-2 positive cell ratios are relatively higher in the HS-ICTO group compared with the HS group, which supports a superior osteoinductive ability of HS-ICTO. Finally, the TRAP staining results demonstrate that the TRAP-positive area is concentrated in proximity to the surface of the HS scaffold, whereas minimal staining is observed in the HS-ICTO group (Supplementary Fig. 79). This finding substantiates the hypothesis that the superior bone regeneration observed in the HS-ICTO group may be partially attributed to the inhibition of osteoclastogenesis. Taken together, these results corroborate that HS-ICTO displays superior osteogenic potential, effectively reinstating the osteogenic-osteoclastic homeostasis and facilitating bone reconstruction in defect sites.

## Discussion

Osteosarcoma is a highly aggressive bone tumor characterized by severe bone destruction, pain, and metastasis. Implants with tumor-killing function have been envisioned as an alternative approach for limb salvage intervention with surgical excision of osteosarcoma. The tumor recurrence of osteosarcoma or inflammatory trauma after limb salvage therapy can lead to a significantly increased level of $H_2O_2$ and hypoxic microenvironments, which may severely disrupt the redox homeostasis and cause cascaded side effects on the osteogenesis process of BMSCs. To address the abovementioned challenges, we have rationally designed and constructed the sono-activable and biocatalytic 3D-printed HS-ICTO scaffold as an intelligent and sequential therapeutic platform in osteosarcoma eradication and bone defect regeneration. The fabricated HS-ICTO exhibits superior POD-like, OXD-like, and GSH-depletion activities under TME conditions for efficient ROS production and also simultaneous CAT-like activity for oxygen supply. After surgical implantation, the HS-ICTO alleviates intratumoral hypoxia and presents highly efficient tumor ablation with sono-activation. Moreover, the intelligently switchable properties enable HS-ICTO to catalyze $H_2O_2$ to $O_2$ for effectively blocking endogenous $H_2O_2$-mediated oxidative stress and supply abundant $O_2$, thereby facilitating the subsequent bone regeneration therapy. The exploitation of such multifunctional bone implants offers a potential on-demand and remote-controlled strategy in response to the challenges of malignant bone tumors.

The HS-ICTO scaffold is designed based on the concept of redox medicine, utilizing $H_2O_2$ as a common therapeutic target for tumor inhibition and the restoration of the osteogenic-osteoclastic balance in bone tissue. The HS-ICTO has been employed as an $H_2O_2$-driven ROS and oxygen-generating system to facilitate different purposes. The hypoxic TME is a primary reason for the insensitivity to ROS-based tumor therapies, as it leads to insufficient ROS generation, failing to disrupt the redox balance within tumor cells. Our results demonstrate that, compared to HS, HS-ICTO significantly alleviates hypoxia in

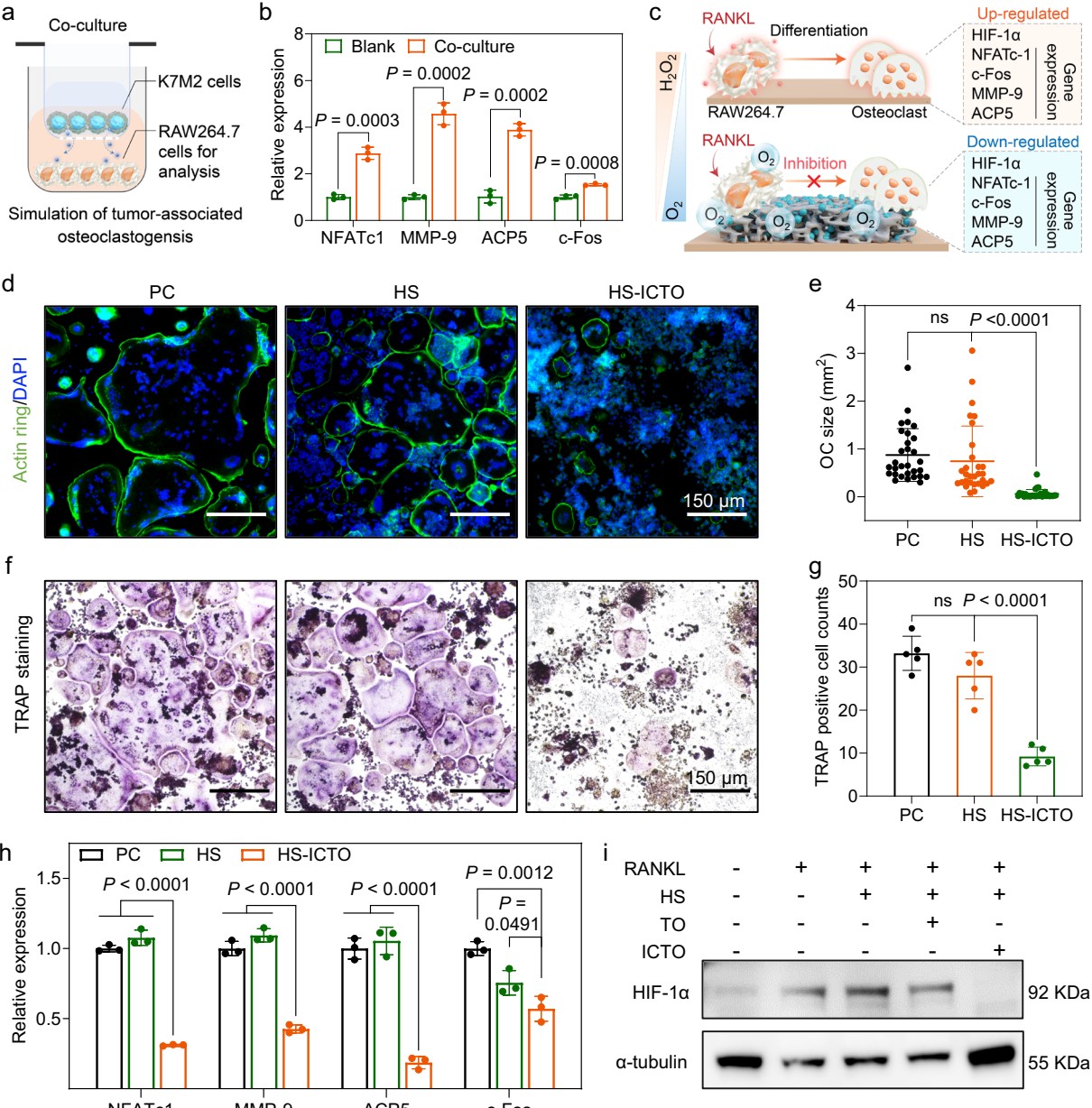

**Fig. 7 | Suppression of osteoclastogenesis of HS-ICTO. a** Illustration of the transwell co-incubation experiment of K7M2 and RAW264.7 cells. The K7M2 cells were seeded in the upper chamber, and the RAW264.7 cells were seeded in the lower chamber. **b** The NFATc1, MMP-9, ACP5, and c-Fos gene expression of RAW264.7 cells co-incubated with K7M2 for 7 days measured by RT-qPCR ($n = 3$ biologically independent replicates). **c** Schematic depiction of the HS-ICTO inhibition mechanism affecting osteoclastogenesis. **d** Representative images of phalloidin-FITC stained RAW264.7 cells after different treatments. The large cells with green actin ring indicate osteoclasts. Positive control group (PC, only stimulated by 50 ng/mL RANKL). **e** Quantitative analysis of the size of osteoclasts ($n = 30$ cellular replicates). **f** Representative images of TRAP-stained RAW264.7 cells after different treatments. The fluorescence images in (**d**, **f**) were representative of three

independently repeated experiments from each group. **g** Quantitative analysis of the TRAP-positive cell counts ($n = 5$ biologically independent replicates). **h** The NFATc1, MMP-9, ACP5, and c-Fos gene expression of RAW264.7 cells after different stimulations for 7 days measured by RT-qPCR ($n = 3$ biologically independent replicates). **i** The expressions of HIF-1α and α-tubulin of RAW264.7 were analyzed by western blot assay. Western blot experiments were independently repeated three times with similar results. Data are presented as mean ± SD, and ns represents no significant difference; in experiment (**b**), statistical significance was calculated using a two-tailed Student's *t*-test and in experiment (**e, g, h**), statistical significance was calculated using one-way ANOVA followed by Tukey's post hoc test for multiple comparisons; all tests were two-sided. Source data are provided as a Source Data file.

tumor tissues and enhances sono-activation of ROS, thereby augmenting antitumor efficiency. In terms of remodeling the damaged bone tissue microenvironment, with CAT-mimetic ability and US withdrawing, the HS-ICTO effectively catalyzes $H_2O_2$ to $O_2$, thereby inhibiting $H_2O_2$-mediated ROS damage and protecting the osteogenic differentiation of BMSCs. Moreover, the enhanced oxygenation also suppresses osteoclastogenesis. Consistent with in vitro findings,

pathology and sequencing data from in vivo models further confirm that HS-ICTO significantly reduces bone resorption and downregulates inflammation-related expression, which eventually offers superior bone regeneration.

The future research directions will focus on exploring the long-term biocompatibility and expanding the clinical applicability and therapeutic scope of the HS-ICTO scaffold. The current study employs

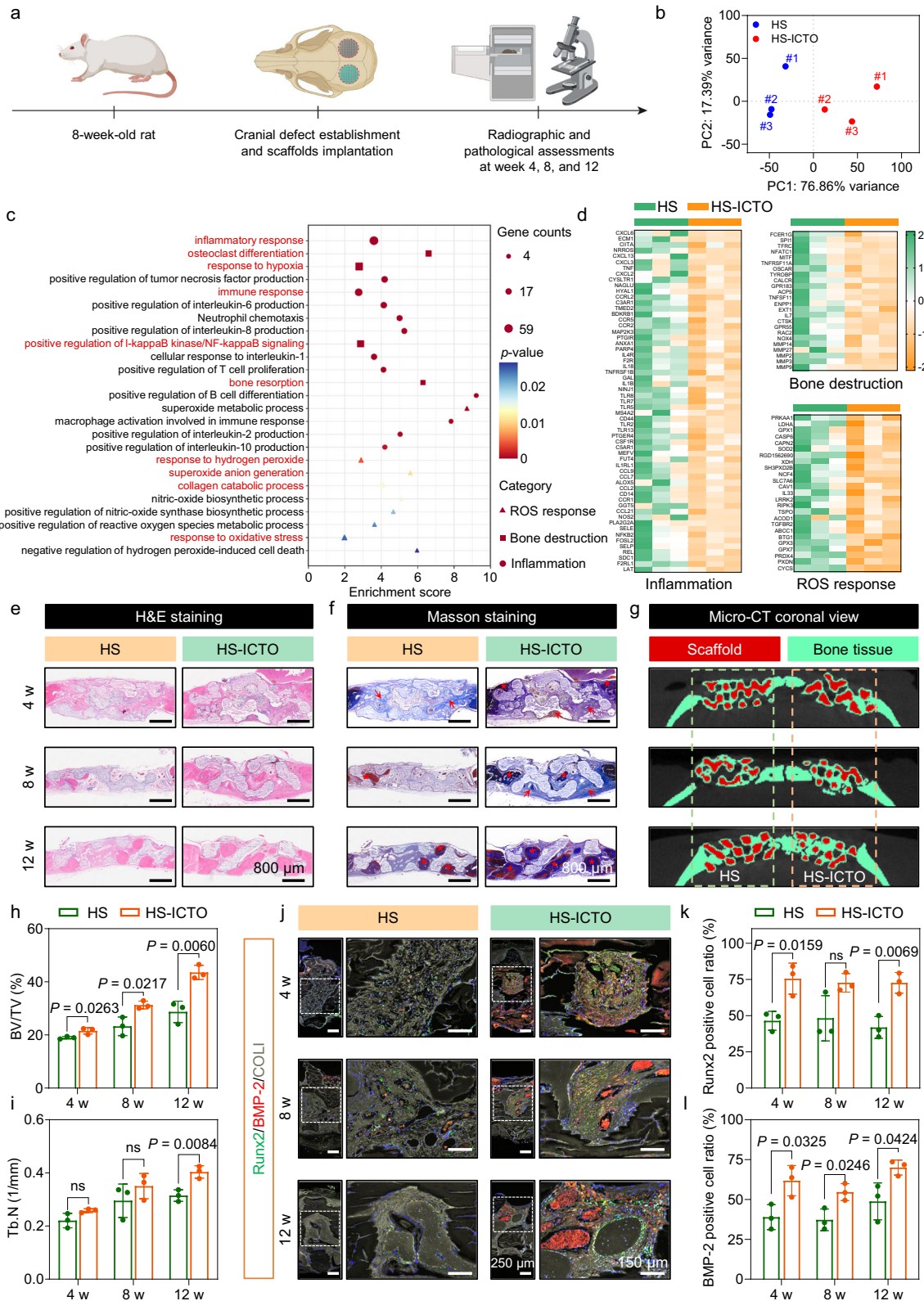

separate animal models of tumor-bearing and bone defects to assess the therapeutic impacts on tumor inhibition and bone regeneration. Our vision includes validating the antitumor and bone repair-promoting capabilities of the HS-ICTO scaffold in an orthotopic osteosarcoma model and a complex large animal bone defect model and aligning these findings with established clinical protocols. Additionally, we aim to conduct comprehensive biological studies to

expedite clinical translation. We envision that HS-ICTO enables rapid tumor clearance at the early stages of recurrence detection following scaffold implantation. Additionally, akin to postoperative radiotherapy or chemotherapy, the ultrasound-activated properties of HS-ICTO facilitate periodic local sono-activated tumor clearance, thus reinforcing the efficacy of the primary surgery and simultaneously minimizing systemic toxicity and adverse effects. We can even expect a scenario

**Fig. 8 | In vivo bone regeneration in rat cranial defect model. a** Process illustration of the in vivo bone regeneration assessments. **b** PCA plots for different samples. **c** GO enrichment analysis of the differentially expressed genes associated with ROS damage, bone destruction, and inflammation in harvested cranial bone tissue. **d** Heat maps illustrating the differential expression profiles. The data in (**b**–**d**) were representative of three biologically independent samples from each group. **e** H&E staining and **f** Masson's trichrome staining of regenerated bones induced by different scaffolds. **g** Coronal views of the defect areas from micro-CT images. **h** Quantitative analysis of bone volume (BV/TV, $n = 3$ biologically independent mice per group) and **i** trabecular number (Tb.N, $n = 3$ biologically independent mice per group) induced by different scaffolds. **j** Immunofluorescence staining of Runx2, BMP-2, and COLI from different scaffolds. The images in (**e**–**g**, **j**) were representative of three biologically independent samples from each group. **k** Quantitative analysis of the Runx2 positive cell numbers ($n = 3$ biologically independent mice per group). **l** Quantitative analysis of the BMP-2 positive cell numbers ($n = 3$ biologically independent mice per group). Data are presented as mean ± SD, and ns represents no significant difference; In (**c**), $P$-values were obtained from one-sided Hypergeometric test without multiple comparisons. The assessment in (**h**, **i**, **k**, **l**) of $P$-values was performed by a two-tailed Student's $t$-test. Source data are provided as a Source Data file.

that combines high-intensity focused ultrasound, a clinically applied method for ablating deep-seated tumors, with HS-ICTO for precise postoperative diagnosis and treatment of osteosarcoma. In general, this study provides a promising strategy for the engineering of multifunctional bone implants with simultaneous preclusion of tumor recurrence and promotion of bone repair based on ROS-catalytic regulation, offering essential guidance for the realization of precise oncological intervention and tissue regeneration.

## Methods

### Synthesis of petaloid protonated titanate (p-PT)
The synthesis of p-PT nanospheres was achieved via a straightforward solvothermal methodology. The reaction medium consisted of a binary solvent system comprising 10 mL of N, N'-dimethylformamide (DMF) and 30 mL of isopropyl alcohol (IPA). Subsequently, 1 mL tetrabutyl orthotitanate (TBT) was introduced into the aforementioned solvent mixture. The reaction mixture was transferred to a 100-mL Teflon-lined autoclave and maintained at 200 °C for a duration of 20 h. Following the completion of the reaction, the p-PT product was isolated via centrifugation at 8000×$g$ and subjected to thorough ethanol washing. The final product was obtained after overnight desiccation at 60 °C.

### Synthesis of petaloid TO
The TO nanospheres were produced through thermal decomposition of the fabricated p-PT petaloid precursor at 450 °C for a period of 2 h in an air atmosphere.

### Synthesis of ICTO
The prepared petaloid TO nanoparticles (100 mg) were dispersed in an ethanol solution (30 mL) containing a predetermined concentration of iridium trichloride (IrCl₃) under continuous agitation for 30 min. Subsequently, the suspension was transferred to a 100-mL Teflon-lined autoclave and subjected to hydrothermal treatment at 180 °C for 12 h. The resultant ICTO was isolated via centrifugation at 8000×$g$ and purified through repeated ethanol washing cycles. The product was then dried at 60 °C overnight.

### Synthesis of HA-based 3D-printed scaffold (HS)
Urethane acrylate (UA, Neorad U25-20D) and poly(ethylene glycol) diacrylate (PEGDA-400) were mixed in a ratio of 3:1 and thoroughly blended with 1 wt% dispersant (BYK-2155, BYK Chemie, Germany). Hydroxyapatite (HAP) at 60 wt% concentration was added to the mixture and milled in a planetary ball mill at 240 r/min for 15 h. After adding 1 wt% photoinitiator (diphenyl(2,4,6-trimethylbenzoyl)-phosphine oxide, TPO), milling was continued for 2 h. The slurry was then separated from the grinding balls and debubbled to obtain the printing slurry. The printing model was designed using Solidworks (Version 2019, Dassault Systemes, France) and imported into Magics (Version 22.0, Materialise, Belgium) to set the printing layer thickness and path. The design model was then transferred to the 3D printer in SLC format for fabrication. After printing, the 3D-printed scaffold was cleaned with anhydrous ethanol and air-dried at room temperature. The 3D scaffold was then sintered in a muffle furnace with a temperature ramping profile: ramped at 5 °C/min to 300 °C and

held for 6 h, ramped at 5 °C/min to 600 °C and held for 2 h, ramped at 5 °C/min to 1150 °C and held for 2 h, and finally cooled to room temperature with the furnace.

### Synthesis of HS-ICTO
A series of HS-ICTO with various concentrations was prepared using a wet deposition method. ICTO was dispersed in ethanol solution at concentrations of 0, 0.5, 1.0, 1.5, and 2.0 mg/mL. The HS scaffold was then immersed in the solution at 37 °C for 1 h. Following this, the scaffold was dried in a constant temperature oven at 85 °C until all the ethanol had evaporated. This process was repeated with the corresponding concentrations of ICTO ethanol solution and dried at 85 °C for a further three times to obtain the final samples.

### Cell line
The 143b human osteosarcoma cell line (Catalog No. CL-0007) and K7M2 murine osteosarcoma cell line (Catalog No. CL-0371) were obtained from Wuhan Pricella Biotechnology Co., Ltd (China). The RAW264.7 cell line (Catalog No. AMC1002) was purchased from Hangzhou Yangming Biotechnology Co., Ltd (China). BMSCs were isolated from neonatal Sprague-Dawley (SD) rats. Briefly, 8-day-old male neonatal rats were euthanized, and their femurs and tibias were aseptically dissected, followed by three washes with PBS containing 2 × penicillin-streptomycin. The proximal and distal metaphysis were carefully removed using sterile scissors to expose the bone marrow cavity. A 1 mL syringe filled with complete growth medium was used to flush the marrow, and the effluent was collected in sterile tubes. After centrifugation at 300×$g$ for 5 min, the pelleted BMSCs were resuspended in complete growth medium and seeded into culture flasks. For cell culture, 143b and K7M2 cells were maintained in RPMI 1640 medium, whereas RAW264.7 cells and BMSCs were cultured in α-DMEM. Both media were supplemented with 10% fetal bovine serum (FBS) and 1% penicillin-streptomycin. All cells were incubated at 37 °C in a humidified atmosphere with 5% $CO_2$.

### Biocompatibility and tumor-killing assays of the scaffolds
To detect the biocompatibility of the scaffolds, the HS-ICTO and HS were sterilized and placed in 48-well plates. Then, the BMSCs were seeded onto the scaffolds at a density of $1 × 10^5$/scaffold with 1 mL culture medium. After 24 and 72 h, the viabilities were tested through CCK-8 assay. Moreover, the cells were stained with calcein-AM for 30 min and propidium iodide for 5 min (C2015M, Beyotime) to demonstrate live/dead cells. After that, the scaffolds were placed in a confocal-exclusive dish and observed by CLSM (N-SIM S, Nikon). As for the tumor-killing assays, the 143b cells the 143b cells were seeded onto the HS-ICTO and HS and incubated for 12 h, allowing cell attachment. Next, US irradiation was applied (1 W/cm², 1.0 MHz, 30% duty cycle, 1 min). After another 12 h, the CCK-8 assay and live/dead staining were performed. The live and dead cell counts were calculated using ImageJ software.

### Animals
All animal experiments and associated procedures (including euthanasia) were conducted in accordance with the animal ethics guidelines

of the Animal Ethics Committee at West China Hospital, Sichuan University, Chengdu, China, under the designated Animal Ethics Committee approval number 20220302062. In this study, 4-week-old female BALB/c nude mice were purchased from GemPharmatech Co., Ltd. (China), and 8-week-old male Sprague-Dawley (SD) rats were purchased from Beijing HFK Bioscience Co., Ltd (China). Both were maintained under standardized laboratory conditions with regulated photoperiods (12:12 h light-dark cycle, light phase: 8:00 a.m. to 8:00 p.m.). The subjects received unrestricted access to standard rodent chow (formulation 1010038, Jiangsu Xietong Pharmaceutical Co., Ltd.) and potable water. Environmental parameters were maintained within specified ranges (ambient temperature: 20–26 °C; relative humidity: 40–70%). Sex and/or gender were not considered in the study design, given that they were not hypothesized to influence the outcomes of this investigation. To minimize animal discomfort and comply with ethical standards, all procedures were strictly followed the Guideline of Assessment for Humane Endpoints in Animal Experiments (Certification and Accreditation Administration of the P.R. China, RB/T 173-2018). According to these regulations, tumor burden should generally not exceed 5% of the animal's normal body weight in standard experiments or 10% of body weight in therapeutic studies. Furthermore, the maximum volume (calculated as (length × width × depth)/2) permitted by the ethics committee for a single tumor is 2000 mm$^3$. All relevant experiments ended once the tumor size reached this limit.

### In vivo osteosarcoma eradication studies
Twenty-five 4-week-old female BALB/c nude mice were randomly divided into five groups ($n = 5$): the sham group, HS group, HS + US group, HS-ICTO group, and HS-ICTO + US group. First, the 143b cells were subcutaneously injected to establish the tumor xenografts (50 μL, $5 × 10^6$ cells per mouse) and allowed to grow for 15 days, which can reach an average tumor volume of approximately 200 mm$^3$. After anesthetizing the mice, the scaffolds (φ = 5 mm × 2 mm) were implanted into the tumor. The US irradiation (Nu-Tek UT1041, 2.5 W/cm$^2$, 1 MHz, 30% duty cycle, 5 min) at the tumor sites was performed on the corresponding group at 24, 48, and 72 h postoperatively. During the treatment, the body weights and tumor volumes were recorded every third day. After 15 days, the mice were euthanized and dissected for histopathological analysis.

### Observation of ALP synthesis and calcium deposition during the osteogenic process of BMSCs
Briefly, BMSCs were seeded in gelatin-coated 24-well plates at a density of $5 × 10^3$ cells per well and allowed to adhere for 12 h. The HS or HS-ICTO was gently placed in the corresponding wells, and the media were replaced by ROS-conditioning or control medium, followed by 24-h incubation. Next, the mediums were replaced by osteogenic medium (α-MEM medium from Gibco containing 10% fetal bovine serum, 100 U/mL penicillin-streptomycin, 50 μg/mL vitamin C, 10 mM β-glycerophosphate, 10 nM dexamethasone) and refreshed every third day. For ALP observation, the cells were fixed on day 14 after osteogenic stimulation, followed by ALP staining using BCIP/NBT Alkaline Phosphatase Color Development Kit (C3206, Beyotime). For calcium deposition analysis, the cells were fixed on day 21 after osteogenic stimulation and stained with Alizarin Red Solution (ALIR-10001, Ori-Cell). The staining results were observed using an inverted fluorescence microscope, and the ALP or ARS positive areas were calculated by ImageJ software.

### Suppression of osteoclastogenesis by HS-ICTO
The co-incubation of K7M2 and RAW264.7 cells was performed in a transwell system. Briefly, the K7M2 cells were seeded in the upper chamber ($2 × 10^4$/well), and RAW264.7 cells were seeded in the lower chamber at a density of $5 × 10^4$/well. On day 5, the mRNA from RAW264.7 cells was extracted, and the expression of NFATc1, MMP-9,

ACP5, and c-Fos was evaluated by RT-qPCR. Primer sequences are shown in Table S4. For the osteoclastogenesis assay, RAW264.7 cells were seeded in 48-well plates ($2.5 × 10^3$). After 12 h, the medium was replaced by osteoclast induction medium (α-MEM medium from Gibco containing 10% heat-inactivated fetal bovine serum, 100 U/mL penicillin-streptomycin, 50 ng/mL RANKL (462-TEC, R&D), and HS, HS-TO, or HS-ICTO extracts). The medium was refreshed every second day until day 7 for the following assays. For actin ring staining, the cells were fixed and permeabilized, followed by Actin-Tracker Green-488 (1:200, C2201S, Beyotime) and DAPI staining. The cells were then observed using an inverted fluorescence microscope, and the cell size was calculated using ImageJ software. The expression of TRAP was evaluated using the TRAP staining kit (Sigma, 387A-1KT) and observed using an inverted fluorescence microscope. The TRAP-positive cell counts were calculated using ImageJ software. The mRNA expression of NFATc1, MMP-9, ACP5, and c-Fos was evaluated by RT-qPCR as mentioned above. The protein expressions of HIF-1α and α-tubulin were evaluated by western blot assay.

### The skull reconstruction in rat cranial defect model
8-week-old male SD rats were anesthetized, and a sterile surgical blade was used to make an incision superior to the sagittal suture to expose the cranial bone. A hand-held trephine drill was utilized to create symmetrical bilateral defects with a diameter of approximately 5 mm. After that, the HS and HS-ICTO (φ = 5 mm) were implanted into the defect area (left: HS; right: HS-ICTO). The implant regions (scaffolds with newly formed bone tissue) were surgically excised at week 4 for RNA-seq analysis ($n = 3$ for HS-ICTO and HS groups). After euthanasia, the cranial bones were collected at weeks 4, 8, and 12 ($n = 3$ for each time point) for radiographic and pathology assessments. Micro-CT was performed using the Quantum GX micro-CT Imaging System. The 3D structures were created using CTVox software, and the coronal view was dyed using Mimics software based on the Hounsfield Unit difference between scaffold and bone tissue. Quantitative analysis of bone mineral density, bone volume versus tissue volume, trabecular number, and trabecular spacing was determined by CTAn software. For pathology analysis, the cranial bones were fixed with 10% formaldehyde and decalcified (E1171, Solarbio). The defective parts were separated and embedded in paraffin. The continuous sliced histological sections were prepared using HM 340E (Thermos) with a thickness of 4 μm. The H&E and Masson-trichrome staining were performed to visualize the new bone formation and collagen deposition. Moreover, Opal staining technology (NEL861001KT, Akoya) was performed to present the expression of collagen I (1:200, ab34710, Abcam), Runx2 (1:300, ab236639, Abcam), and BMP-2 (1:100, ab214821, Abcam). The positive cell ratios were evaluated using Qupath software. TRAP staining (387A-1KT, Sigma) was performed to reveal the osteoclast of the tissue sample, and the positively stained area was calculated using ImageJ software.

### Statistical analysis
All tumor sizes and mice body weights were recorded using Microsoft Office 2019. GraphPad Prism Version 9.5 software (GraphPad Prism, San Diego, California, USA) was used to analyze statistical data. FlowJo Version 10.8.1 was used to analyze flow cytometry data. ImageJ Version 1.52 v and Qupath 0.4.0 were used to analyze immunofluorescent and immunochemical data. Figures were formed using Origin 2024. Sample size ($n$), probability ($P$) value, data normalization, and specific statistical tests for each experiment were clarified in the figure legends. All data were presented as the mean ± standard deviation (SD). The one-way analysis of variance (ANOVA) with Tukey's post hoc test was employed to analyze differences among multiple groups, and a two-tailed Student's $t$-test was performed for the comparison between the two groups. All tests were two-sided. A value of $P < 0.05$ was used to indicate statistical significance.

## Figures and artwork

Graphic elements in Figs. 1, 2a, 3a, 4a, 5a, b, 6a, n, 7a, c, 8a, n, and Supplementary Figs. 1a, 2a, 57b, 68b were created using the open-source software Blender 3.4 and Inkscape 1.4.2, both distributed under the GNU General Public License (GPL).

## Reporting summary

Further information on research design is available in the Nature Portfolio Reporting Summary linked to this article.

## Data availability

The main data supporting the results of this study are available within the paper and its Supplementary Information. Any other raw data or noncommercial material used in this study are available from the corresponding author. Raw RNA sequencing data generated in this study have been deposited in the NCBI SRA database under accession number GSE296989. Source data are provided with this paper.

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

## Acknowledgements

This work was financially supported by the National Key Research and Development Program of China (2024YFE0201200 [C.C.]), National Natural Science Foundations of China (82202187 [X.R.], 82272003 [L.Q.], 52373148 [C.C.], 52173133 [C.C.]), Sichuan Science and Technology Program (2024YFHZ0271 [X.R.]), Fundamental Research Funds for the Central Universities [C.C.]. We gratefully acknowledge Dr. Mi Zhou and Dr. Chao He for their analytical support, and we also thank Li Li, Fei Chen, and Chunjuan Bao of the Institute of Clinical Pathology, Sichuan University, for processing histological staining. We are grateful to Hanjiao Chen of the Analytical & Testing Center of Sichuan University for technical assistance with EPR, and we also thank the Shanling Wang of the Analytical & Testing Center of Sichuan University for the assistance with HAADF-STEM imaging. Furthermore, we extend our gratitude to Diwei Wu of the Orthopedic Research Institute, Department of Orthopedics, West China Hospital, Sichuan University, for providing the experimental protocols and operational guidance for some of the cell experiments.

## Author contributions

X.R. and S.T.X. contributed equally to this work. X.R., S.T.X., L.Q., and C.C. conceived the idea and designed the project. X.R. and S.T.X. performed the experiments and analyzed the results. W.G. and S.T.X. designed and conducted the DFT theoretical calculations. B.H.Z., P.M., Z.C.D., B.Q.Z., and Y.J.F. assisted with the figure production and experimental design. X.R., S.T.X., L.Q., and C.C. wrote the manuscript. B.Q.Z., L.Q., and C.C. corrected the manuscript and supervised the whole project. All authors discussed the results and commented on the manuscript.

## Competing interests

The authors declare no competing interests.
