## [Transparent Peer Review file · Nature Communications]

Sono-Activable and Biocatalytic 3D-Printed Scaffolds for Intelligently Sequential Therapies in Osteosarcoma Eradication and Defect Regeneration

Corresponding Author: Professor Chong Cheng

Version 0:

Reviewer comments:

Reviewer #1

(Remarks to the Author)

This study introduces a sono-activable and biocatalytic 3D-printed scaffold with paradoxical yet synergistic functions, designed to enable sequential therapies for osteosarcoma eradication and defect regeneration following surgical implantation. The scaffold employs a multifunctional enzyme-like biocatalytic process to adaptively regulate the microenvironment, suppressing osteosarcoma proliferation while fostering bone tissue regeneration. The exploitation of such multifunctional bone implants offers a potential on-demand and remote-controlled strategy in response to the challenges of malignant bone tumors. Overall, the concepts presented in this study are interesting. I recommend its publication after the following aspects are addressed:

- (1) The authors propose that the fabricated ICTO nanoparticles and HS-ICTO scaffolds exhibit versatile H₂O₂-catalytic performance. However, the underlying reasons for their high H₂O₂ substrate affinity require further exploration. Additional analysis and discussion on this aspect would enhance the depth of the study.
- (2) While the authors have validated the multifunctional ROS catalytic activities of ICTO, the enzyme-like catalytic performance after loading onto the scaffold appears to be incomplete. Providing a more detailed explanation and further clarification of this limitation would strengthen the manuscript.
- (3) Although these biocatalytic scaffolds exhibit high H₂O₂-catalytic performance, it remains unclear whether these materials can maintain stable functionality over extended periods. Specifically, an evaluation of the long-term structural and biocatalytic stability of the ICTO nanoparticles and HS-ICTO scaffolds, such as their H₂O₂-catalytic stability over a 2-3 week period, would strengthen the discussion.
- (4) What are the main advantages of the HS-ICTO scaffolds compared to other reported ROS-scavenging scaffolds, such as those modified with catechol chemistry? A more in-depth discussion on this comparison would highlight the essential findings and unique contributions of this study.
- (5) What are the interactions between hydroxyapatite (HA) 3D-printed scaffolds and ICTO nanoparticles? Are these interactions stable, particularly within a body fluid environment.
- (6) To strengthen the demonstration of biocompatibility, it is recommended to include comprehensive blood routine test data in Supplementary Figure 48.
- (7) On page 13, could you provide a more detailed explanation of how the cytotoxicity of ICTO against 143b osteosarcoma cells and BMSCs in vitro is affected by specific tumor microenvironment (TME) conditions?
- (8) In the osteosarcoma model, how is the size of the 3D-printed scaffold determined and how is the concentration of nanoparticles optimized?
- (9) After tumor treatment, can the scaffold degrade naturally, or does it require surgical removal?
- (10) Hydroxyapatite (HA) exhibits certain piezoelectric properties. In the osteosarcoma model, the scaffold alone has been observed to inhibit tumor growth. Could this effect be potentially related to its piezoelectric characteristics?
- (11) The inclusion of staining images for Caspase-3 in vivo is recommended, as they are crucial for confirming the apoptosis process in vivo.
- (12) On page 26, it is essential to include appropriate references to support the discussion on differential gene expression following HS-ICTO treatment.

Reviewer #2

(Remarks to the Author)

This manuscript presents an interesting 3D-printed scaffold with sono-activable and biocatalytic properties for intelligently sequential therapies in osteosarcoma eradication and defect regeneration after surgical implantation. Such therapeutic systems display superior, spatiotemporally controllable, and versatile H₂O₂-catalytic performances, which promptly generates ROS in TME via multienzyme-like pathways in conjunction with sono-activation. Concurrently, it can intelligently switch to catalyze H₂O₂ to O₂ for effectively blocking endogenous H₂O₂-mediated oxidative stress during the subsequent bone defect regeneration. This research provides a new and promising strategy for the engineering of multifunctional bone implants with simultaneous preclusion of tumor recurrence and promotion of bone repair based on ROS-catalytic regulation, offering essential guidance for the realization of precise oncological intervention and tissue regeneration. The results are innovative and the manuscript is well-structured, containing extensive and comprehensive studies and convincing experimental evidences. Overall, I would like to recommend the publication of the work in Nature Communications after addressing the following issues:

1. For the rationale for constructing biocatalysts with multi-edge petaloid microstructure, more explanation is needed. For example, what specific advantages does this structural characteristic confer to their performance or application?
2. In Fig. 1j, why was Ir/C selected as a reference in the Ir 4f analysis, and what insights does it provide into the electronic structure of the Ir site in ICTO? This is not clearly explained.
3. For the mechanism of intelligent performance conversion of ICTO, it is necessary to further clarify the relationship between it and biological applications.
4. Fig. 5f and supplementary Fig. 45 illustrate the endocytosis of ICTO nanoparticles. The authors may discuss whether this process poses a risk of cumulative metal ion toxicity.
5. Please provide more comments or experiments to clarify whether the mechanical strength of the prepared 3D scaffold is sufficient to fill large segmental bone defects.
6. Page 20, the authors proposed, "...the excessive H₂O₂ and insufficient O₂ supply may lead to cascaded side effects on the osteogenesis process of BMSCs by restraining motility..." It is necessary to supplement cell migration-related experiments to evaluate the ability of HS-ICTO to recruit stem cells.
7. Osteosarcoma typically occurs in large weight-bearing bones such as the femur, but the authors choose a cranial bone defect model to simulate postoperative bone defect repair instead of a weight-bearing bone defect model. So the authors are suggested to provide the connection of these animal models.

Version 1:

Reviewer comments:

Reviewer #1

(Remarks to the Author)

The revised manuscript is recommended for publication.

Reviewer #2

(Remarks to the Author)

The authors have addressed all my concerns, so I suggest its rapid publication.

Point-by-point response to the detailed comments by the reviewers of “*Sono-Activable and Biocatalytic 3D-Printed Scaffolds for Intelligently Sequential Therapies in Osteosarcoma Eradication and Defect Regeneration*”

REVIEWER COMMENTS

Reviewer #1:

General comment: *“This study introduces a sono-activable and biocatalytic 3D-printed scaffold with paradoxical yet synergistic functions, designed to enable sequential therapies for osteosarcoma eradication and defect regeneration following surgical implantation. The scaffold employs a multifunctional enzyme-like biocatalytic process to adaptively regulate the microenvironment, suppressing osteosarcoma proliferation while fostering bone tissue regeneration. The exploitation of such multifunctional bone implants offers a potential on-demand and remote-controlled strategy in response to the challenges of malignant bone tumors. Overall, the concepts presented in this study are interesting. I recommend its publication after the following aspects are addressed.”*

Response to the general comment:

Thanks for your constructive comments and advice on how to improve the quality of this research article. We have thoroughly and carefully corrected the manuscript based on your suggestion, and all the necessary data have been added to support our claims. Meanwhile, all the questions and concerns have been carefully addressed in the revised manuscript, as well as supplementary information. Thus, we believe that the quality of this paper has been significantly enhanced. Again, we sincerely thank you for your great efforts.

(1) *“The authors propose that the fabricated ICTO nanoparticles and HS-ICTO scaffolds exhibit versatile H₂O₂-catalytic performance. However, the underlying reasons for their high H₂O₂ substrate affinity require further exploration. Additional analysis and discussion on this aspect would enhance the depth of the study.”*

Response to this comment:

We appreciate your constructive comment. ICTO functions as the pivotal active component within the composite scaffold, conferring exceptional H₂O₂ affinity to the HS-ICTO. Accordingly, we

will systematically elucidate the mechanism underlying the high H₂O₂ substrate affinity exhibited by ICTO nanoparticles, which can be summarized in three aspects. First, Ir redox-active centers of ICTO, characterized by partially occupied 5*d* orbital configurations, facilitate optimal overlap with the electronic structure of H₂O₂, thus weakening the O-O bond for activation and accelerating the subsequent formation of reaction intermediates during biocatalysis. Additionally, the redox properties of Ir, manifested through multiple stable oxidative states, enable efficient electron/proton transfer processes, thereby enhancing H₂O₂ adsorption and activation efficiency (*Nat. Commun.* **12**, 6118 (2021), *J. Am. Chem. Soc.* **146**, 6846–6855 (2024), *Sci. Adv.* **9**, eadi8025 (2023)). Second, by constructing Ti-O-Ir structures with potent chemical coupling on biocompatible and physiologically stable TiO₂ nanocrystalline surfaces, precise modulation of Ir active site characteristics was achieved with rationally optimized size and dispersion of Ir clusters, thereby exposing more surface area and providing sufficient active sites to enhance H₂O₂ interaction (*Angew. Chem. Int. Ed.* **63**, e202406427 (2024), *Nat. Commun.* **15**, 3765 (2024)). Third, the hierarchically structured and high-surface-area morphology of the petaloid microsphere promotes sufficient exposure of Ir active sites and H₂O₂ accessibility, thereby enhancing reaction kinetics and catalytic efficiency.

Furthermore, the superior substrate affinity was corroborated through *in vitro* enzyme-like biocatalytic activity tests, where ICTO exhibited significantly lower *K_m* values during H₂O₂ biocatalysis (Figs. 2c, g and Supplementary Table 1). Meanwhile, Gibbs free energy analysis of density functional theory (DFT) calculations suggested that the adsorption and self-decomposition of H₂O₂ in the initial stage of the reaction were remarkably exothermic processes (Fig. R1). Collectively, benefiting from the distinctive morphological features and highly exposed Ir active sites of ICTO, it demonstrates excellent H₂O₂ substrate affinity.

We hope this detailed explanation addresses your question satisfactorily. Thank you again for your insightful comments and for helping us improve the clarity and depth of our manuscript.

Fig. 2 c Comparison of kinetic parameters in ICTO and TO. **g** A comparative analysis of the V_{max} and K_m parameters for ICTO and TO.

Fig. R1. Calculated the free energy for the adsorption of H_2O_2 by ICTO.

(2) “While the authors have validated the multifunctional ROS catalytic activities of ICTO, the enzyme-like catalytic performance after loading onto the scaffold appears to be incomplete. Providing a more detailed explanation and further clarification of this limitation would strengthen the manuscript.”

Response to this comment:

We sincerely thank the reviewer for their valuable feedback regarding the enzyme-like catalytic performances of ICTO after being loaded onto the scaffold, and we acknowledge the limitations in our initial characterization of the multifunctional ROS-catalytic activities inherent to the HS-ICTO composite scaffolds. To further improve and strengthen our research validation, we have added supplementary experiments, including pH-dependent ROS generation activity, GSH depletion assays,

sono-activated ROS amplification, and O₂ generation studies of HS-ICTO. The resultant experimental data has been added to the revised supplementary information with interpretations integrated into the revised manuscript as detailed as follows:

Page 13 in the revised manuscript: “Fig. 3a illustrates the proposed therapeutic mechanism of HS-ICTO, which involves the conversion of H₂O₂ to •O₂⁻ (POD-like behavior) and O₂ (CAT-like behavior) under TME conditions, coupled with sono-activation effects to generate substantial ROS in the TME rapidly. Concurrent GSH catabolism synergistically disrupts cellular redox homeostasis. The observed multifaceted enzyme-like activities and ultrasound-amplifying effects of HS-ICTO align with those of bare ICTO (Figs. 3b, c, and Supplementary Figs. 38-44).”

Supplementary Fig. 39. a UV-vis absorption spectra of POD-like activity of HS-ICTO-x. **b** Quantitative analysis of POD-like activity of HS-ICTO-2.0 at different pH conditions (n = 3 independent experiments, data are presented as mean ± SD). Source data are provided as a Source Data file.

Supplementary Fig. 41. a GSH depletion ability of HS-ICTO and HS with DTNB as the trapping agent of -SH in GSH. **b** Quantitative analysis of GSH consumption ratios of HS-ICTO and HS ($n = 3$ independent experiments, data are presented as mean \pm SD). The assessment of P -values is performed by a two-tailed Student's t -test. Source data are provided as a Source Data file.

Supplementary Fig. 42. a The spectra of $\bullet\text{O}_2^-$ generation and sonication enhancement after co-culture with HS-ICTO detected by HE probes. **b** Relative amount of $\bullet\text{O}_2^-$ generation after co-culture with various treatment groups ($n = 3$ independent experiments, data are presented as mean \pm SD; US: 2.0 W/cm^2 , 1 MHz, 30% duty cycle). Statistical significance was calculated using one-way ANOVA followed by Tukey's post-hoc test for multiple comparisons, and ns represents no significant difference. Source data are provided as a Source Data file.

Supplementary Fig. 43. a Sono-stimulated catalytic oxidation of DPA after co-culture with HS-ICTO. **b** Relative activity of $^1\text{O}_2$ generation after sono-activation ($n = 3$ independent experiments, data are presented as mean \pm SD; US: 2.0 W/cm², 1 MHz, 30% duty cycle). The assessment of P -values is performed by a two-tailed Student's t -test. Source data are provided as a Source Data file.

Supplementary Fig. 44. Real-time O₂ generation curve of HS-ICTO in the presence of H₂O₂ at pH=6.5. Source data are provided as a Source Data file.

(3) “Although these biocatalytic scaffolds exhibit high H₂O₂-catalytic performance, it remains unclear whether these materials can maintain stable functionality over extended periods. Specifically, an evaluation of the long-term structural and biocatalytic stability of the ICTO nanoparticles and HS-ICTO scaffolds, such as their H₂O₂-catalytic stability over a 2-3 week period, would strengthen the discussion.”

Response to this comment:

Thank you for your invaluable comments and suggestions on how to enhance the quality of our manuscript. In response to your recommendations, we have conducted the *in vitro* long-term stability tests to demonstrate that ICTO nanoparticles and HS-ICTO scaffolds can maintain sustained, efficient, and stable H₂O₂-catalytic capabilities. Given the distinct pathophysiological conditions of osteosarcoma and cranial defects, we carried out stability assessments in phosphate-buffered saline (PBS) solutions at pH 6.5 and 7.4 to simulate the tumor microenvironment and bone defect condition, respectively. Specifically, ICTO and HS-ICTO were immersed in PBS at both pH levels for extended periods, with H₂O₂-catalytic activity (including POD-like and CAT-like activities) assessed at predetermined time points. As shown in Supplementary Figs. 28, 45, even after four weeks of immersion under acidic conditions (pH 6.5), ICTO can maintain its characteristic multi-edge petaloid morphology, and both the ICTO and HS-ICTO scaffolds exhibit superior ROS generation performance without noticeable deterioration. Additionally, under physiological conditions (pH 7.4), the scaffolds displayed virtually undiminished H₂O₂ decomposition efficiency throughout four weeks (Supplementary Fig. 64). These findings indicate that both ICTO nanoparticles and HS-ICTO 3D-printed scaffold possess sustained H₂O₂-catalytic performance and structural integrity over extended periods.

In general, the inflammatory phase predominantly manifests during the initial stages of bone defect repair (*Bone Res.* **9**, 18 (2021), *Adv. Funct. Mater.* **33**, 2212738 (2023)). Our observations indicate that the surgical implantation of HS-ICTO at the defect area effectively modulates severe inflammation. During the proliferation and microenvironment remodeling phases of bone repair, as inflammation gradually subsides, H₂O₂ expression levels normalize, suggesting the redox microenvironment nearly achieves stability, and it is believed that anti-inflammatory utility may not be required (*Nat. Rev. Mol. Cell Biol.* **21**, 696–711 (2020)). Therefore, we posit that the four-week H₂O₂ catalysis stability tests adequately demonstrate the sustained therapeutic efficacy throughout the bone defect treatment process. We acknowledge these conjectures merit further investigation, and we plan to develop appropriate tracking methodologies in future studies to better understand the behavior and interactions during long-term tissue regeneration of HS-ICTO scaffolds. Again, we appreciate your valuable suggestions, which will undoubtedly guide our future research. The corresponding details have been incorporated into the revised manuscript and supplementary information as indicated:

Page 10 in the revised manuscript: “Furthermore, we evaluated the long-term activity and stability of ICTO, which reveal the consistent preservation of its morphology and sustained high-efficiency in H₂O₂ biocatalysis without appreciable deterioration (Supplementary Fig. 28).”

Page 14 in the revised manuscript: “Besides, extended durability analyses demonstrate that HS-ICTO maintains continuous, efficient, and stable ROS generation capacity, further highlighting its remarkable potential for prolonged therapeutic applications (Supplementary Fig. 45).”

Page 20 in the revised manuscript: “Subsequent investigations into the temporal stability of HS-ICTO’s H₂O₂ decomposition catalysis reveal consistent activity with negligible performance variation even after four-week exposure to simulated physiological conditions, emphasizing its viability for sustained therapeutic interventions (Supplementary Fig. 64).”

Morphostructural stability of ICTO nanoparticles

Stability of H₂O₂-catalytic performance of ICTO nanoparticles

Supplementary Fig. 28. The morphological structure and H₂O₂-catalytic performance stability of ICTO nanoparticles were determined at different immersion time points. **a** SEM images of ICTO following the 7, 14, 21, and 28-day immersion period. **b** POD-like activity (n = 3 independent experiments). **c** CAT-like activity (n = 3 independent experiments). Data are presented as mean ± SD, and ns represents no significant difference; statistical significance was calculated using one-way ANOVA followed by Tukey's post-hoc test for multiple comparisons. Source data are provided as a Source Data file.

Supplementary Fig. 45. POD-like activity measurement of HS-ICTO after immersion for different times. Data are presented as mean ± SD, and ns represents no significant difference; statistical significance was calculated using one-way ANOVA followed by Tukey's post-hoc test for multiple comparisons. Source data are provided as a Source Data file.

Supplementary Fig. 64. CAT-like activity measurement of HS-ICTO after immersion for different times. Data are presented as mean \pm SD, and ns represents no significant difference; statistical significance was calculated using one-way ANOVA followed by Tukey's post-hoc test for multiple comparisons. Source data are provided as a Source Data file.

(4) *“What are the main advantages of the HS-ICTO scaffolds compared to other reported ROS-scavenging scaffolds, such as those modified with catechol chemistry? A more in-depth discussion on this comparison would highlight the essential findings and unique contributions of this study.”*

Response to this comment:

Thank you for your insightful feedback. We will clarify and emphasize the unique advantages of our fabricated HS-ICTO scaffolds in terms of catalytic performance, stability, and performance-intelligent switching for sequential therapies in osteosarcoma eradication and defect regeneration after surgical implantation.

1) Enhanced catalytic efficiency and availability of active site: The HS-ICTO scaffolds ensure superior substrate accessibility and optimal exposure of active components by strategically depositing numerous ICTO nanoparticles with high specific surface area and abundant Ir clusters-based redox active sites, thereby exhibiting exceptional and long-term H₂O₂-catalytic performance. In comparison, catechol-modified scaffolds primarily rely on the intrinsic antioxidant properties of surface-bound ortho-diphenol functional groups, which may display suboptimal performance due to irreversible oxidation reaction and surface saturation, ultimately reducing their overall reaction kinetics and long-term activity in practical applications.

2) Sustained and controllable activity: As addressed in our previous response, the HS-ICTO scaffold demonstrates long-term, efficient, and stable H₂O₂-catalytic capability. Furthermore, its sono-activable effects under TME conditions enable controllable modulation of ROS generation. Conversely, catechol-modified scaffolds primarily achieve their antioxidant effects through reduction reactions, experiencing degradation and subsequent diminishment in ROS scavenging capacity due to ROS oxidation. Consequently, these limitations render them inappropriate for sustained therapeutic applications demanding controlled ROS modulation.

3) Intelligent performance switching in specific pathological microenvironments: The HS-ICTO scaffold possesses multienzyme-like properties and unique dynamic ROS-catalytic switching

capabilities, adapting its functionality according to the local characteristic microenvironments. Specifically, under TME conditions, it exhibits exceptional POD-like, OXD-like, and GSH-depletion activities with sono-activable properties, which rapidly generates substantial ROS for efficient tumor ablation. During subsequent bone defect regeneration in neutral pH condition, the intelligently switchable biocatalytic properties from ROS generation to ROS scavenging enable HS-ICTO to catalyze H_2O_2 to O_2 for effectively blocking endogenous H_2O_2 -mediated oxidative stress and maintaining optimal oxygenation for osteogenesis. In contrast, catechol-modified scaffolds are limited to ROS scavenging properties, which can't achieve ROS generation when changing the pH in microenvironment.

We hope this response adequately addresses your concerns and highlights the distinct advantages of our synthesized HS-ICTO scaffolds. Thanks again for your valuable comments and suggestions.

(5) *“What are the interactions between hydroxyapatite (HA) 3D-printed scaffolds and ICTO nanoparticles? Are these interactions stable, particularly within a body fluid environment.”*

Response to this comment:

We sincerely appreciate your insightful and constructive comments aimed at enhancing the quality of our manuscript. The wet evaporation and deposition strategy has been employed in this study, which mainly involves the precipitation and adsorption of ICTO nanoparticles on the surface of 3D-printed hydroxyapatite (HA) substrates. The interfacial effects in this process are predominantly characterized by physical interactions, encompassing roughness effects and surface adsorption forces. Specifically, the 3D printing process endows the HA scaffold with a porous architecture, enhancing the specific surface area and generating micron-scale surface roughness. Meanwhile, ICTO nanoparticles exhibit multi-edged petaloid micromorphology, significantly increasing their inherent surface roughness. This dual structural feature allows ICTO nanoparticles to effectively anchor on the surface porous structures of the scaffold. Additionally, the van der Waals forces and hydrogen bonds between ICTO nanoparticles and the 3D-printed HA scaffolds further augment the anchoring effects of ICTO. This multifaceted interaction mechanism ensures robust stability of ICTO nanoparticles when depositing on the HA scaffold surface, which enables them to withstand scouring, washing, and long-term soaking, with detachment occurring only under mechanical scraping or energy disturbance.

To substantiate the aforementioned hypothesis, we immersed the scaffolds in PBS for an extended

period to simulate the physiological microenvironment. At predetermined intervals, we collected the solution and employed UV-Vis-NIR spectroscopy to monitor absorbance changes, thereby determining the release quantity of ICTO nanoparticles. Spectroscopic analysis revealed negligible changes in transmittance even after prolonged immersion in the PBS, suggesting robust interfacial stability between the ICTO nanoparticles and the HS scaffold. Notably, upon the application of ultrasound, there was a certain degree of reduction in solution transmittance, demonstrating the controlled release of ICTO nanoparticles through ultrasonic energy modulation (Supplementary Fig. 4). This behavior also facilitated the sono-activated intracellular uptake of ICTO during the antitumor process, potentially augmenting the antitumor efficacy. The corresponding data has been incorporated into Supplementary Fig. 4 of the revised supplementary information, with experimental procedures added to the supplementary methods section. The relevant findings have been discussed in the revised manuscript as follows:

Page 6 in the revised manuscript: “Notably, the petaloid morphology and hierarchical porous structure of ICTO, in conjunction with the porous structures and rough surface characteristics of 3D-printed HS, facilitate the robust interfacial adhesion and anchoring of ICTO. This interaction ensures the relatively good stability for ICTO nanoparticles on the HS substrate, which can withstand the scouring, washing, and long-term soaking. Nanoparticle detachment is observed exclusively under mechanical scraping or external energy disturbance (Supplementary Fig. 4).”

Page 5 in the revised supplementary information: “To elucidate the robust interfacial interaction between ICTO nanoparticles and HS within the fabricated HS-ICTO scaffold, we subjected the scaffold to prolonged immersion in phosphate-buffered saline (PBS) to simulate the physiological microenvironment. The experimental results reveal that ICTO nanoparticles maintain stable adhesion to the surface of HS scaffold throughout the extended immersion period, with negligible reduction in the transmittance of the immersion solution. Upon sonication-induced mechanical stress, while some particle detachment occurred, over 60% of ICTO nanoparticles remain anchored to the scaffold, validating the durability of the fabricated HS-ICTO scaffolds.”

Page 57 in the revised supplementary information: “**ICTO nanoparticle release experiments.** Release kinetics study of ICTO nanoparticles was conducted using UV-Vis-NIR spectroscopy to evaluate the transmittance of solutions at various immersion time points. The prepared HS-ICTO scaffolds were immersed in 4 mL PBS buffer, and the immersion solutions were collected at

predetermined time intervals (7, 14, 21, 28-day). The transmittance of the solutions was measured using UV-Vis-NIR spectroscopy. The release quantity of ICTO nanoparticles was calculated using pure PBS as a reference.”

Supplementary Fig. 4. a Transmittance spectra of solutions with different immersion times in the visible wavelength range. **b** Quantitative analysis of transmittance in the immersion solution across the 350-600nm wavelength range and retention rate of ICTO nanoparticles on the scaffold (n = 3 independent experiments, data are presented as mean ± SD; US *in vitro*: 1.0 W/cm², 1 MHz, 30% duty cycle, 5 min; US *in vivo*: 2.5 W/cm², 1 MHz, 30% duty cycle, 5 min). Source data are provided as a Source Data file.

(6) “To strengthen the demonstration of biocompatibility, it is recommended to include comprehensive blood routine test data in Supplementary Figure 48.”

Response to this comment:

We extend our sincere gratitude for your insightful suggestions and meticulous considerations regarding the biosafety of the scaffolds. To address this issue, we conducted subcutaneous implantation of HS and HS-ICTO in the dorsal region of 8-week-old male Sprague-Dawley (SD) rats. Peripheral blood was collected after 4 weeks for comprehensive hematological and biochemical assessments of liver and kidney functions (Supplementary Fig. 61). The results demonstrated that essential hematological parameters (RBC, HGB, PLT, and WBC) in both groups remained within the normal physiological range, with no significant inflammatory or anemic alterations observed. Additionally, biochemical assays revealed no notable abnormalities in critical indicators such as ALT, AST, UREA, and CREA, thereby substantiating the circulatory safety of HS and HS-ICTO over extended metabolic

periods. The relevant findings have been discussed in the revised manuscript as follows:

Page 17 in the revised manuscript: “Moreover, to further underscore the biosafety profile of HS-ICTO, we conducted subcutaneous implantation of HS and HS-ICTO in the dorsal region of 8-week-old male Sprague-Dawley (SD) rats. Peripheral blood was collected after 4 weeks for comprehensive hematological and biochemical evaluations of liver and kidney functions (Supplementary Fig. 61). The results demonstrated that parameters including RBC, HGB, PLT, and WBC in both groups remained within the normal physiological range, with no significant inflammatory or anemic alterations observed. Furthermore, liver/kidney function biochemical assays revealed no notable abnormalities in critical indicators such as ALT, AST, UREA, and CREA, thereby substantiating the circulatory safety of HS and HS-ICTO over extended metabolic periods.”

Supplementary Fig. 61. a Blood routine test results including red blood cell count (RBC), hemoglobin (HGB), platelets (PLT), and white blood cell count (WBC). **b** Biochemistry test results including alanine aminotransferase (ALT), aspartate aminotransferase (AST), urea, and creatinine (CREA) (n = 3 biologically independent replicates). Data are presented as mean \pm SD, and ns represents no significant difference; statistical significance was calculated using two-tailed Student’s *t*-test. Source

data are provided as a Source Data file.

(7) “On page 13, could you provide a more detailed explanation of how the cytotoxicity of ICTO against 143b osteosarcoma cells and BMSCs in vitro is affected by specific tumor microenvironment (TME) conditions?”

Response to this comment:

Thanks for your valuable and constructive feedback on enhancing the quality of this manuscript, and we apologize for not clarifying this statement earlier. In response to your comment, we have elaborated on how specific TME conditions affect the cytotoxicity of ICTO on 143b osteosarcoma cells and BMSCs and have incorporated the relevant explanations in the revised manuscript.

During the pathological progression of osteosarcoma, malignant tumor cells activate the Warburg effect through abnormally rapid proliferation, primarily relying on anaerobic glycolysis for energy metabolism. This metabolic reprogramming not only leads to a significant accumulation of lactic acid, contributing to the characteristic acidic nature of the tumor microenvironment but also induces excessive expression of H_2O_2 , disrupting the redox homeostasis of the TME, which is a defining characteristic of TME (*Natl. Sci. Rev.* **5**, 269-286 (2018)). According to our experimental studies and theoretical calculations, ICTO enables superior and pH-dependent ROS generation activity, primarily producing ROS under mildly acidic conditions, with no ROS effects observed under neutral conditions (Supplementary Figs. 14, 15). Furthermore, this has been validated at the cellular level. Specifically, at an ICTO concentration of 150 $\mu\text{g/mL}$, its cytotoxicity to BMSCs is negative (viability >95%), whereas the survival rates of 143b osteosarcoma cells at this concentration are 67.3% (pH 7.0 medium) and 45.3% (pH 6.5 medium to simulate TME conditions), respectively (Supplementary Fig. 35). This indicates that ICTO can generate substantial ROS in tumor cells, inducing overloaded oxidative stress in 143b osteosarcoma cells, and ultimately leading to cancer cell death, thereby implying the relevant of cytotoxicity to specific TME conditions.

We hope that the above explanations have answered your questions and would like to thank you again for your comments and suggestions. The relevant statements have been supplemented in the revised manuscripts, which are also shown as follows:

Page 13 in the revised manuscript, “**Firstly, the biocompatibility of bare ICTO nanoparticles on bone stem cells was carried out using the Cell-Counting-Kit-8 (CCK-8) assay at a predetermined**

concentration. As illustrated in Supplementary Fig. 35, both ICTO and TO nanoparticles demonstrate negligible cytotoxicity towards primary-cultured rat bone marrow mesenchymal stem cells (BMSCs, pH 7.4 medium) at concentrations up to 150 $\mu\text{g}/\text{mL}$, maintaining cellular viability above 95%. Conversely, exposure of 143b osteosarcoma cells to 150 $\mu\text{g}/\text{mL}$ ICTO under simulated TME conditions (pH 6.5 medium) results in a significant reduction of cell viability to 45.3%, suggesting that ICTO can specifically respond to the acidic microenvironment of tumor cells, thereby inducing oxidative stress overload and apoptosis of tumor cells.”

Supplementary Fig. 14. **a** UV-vis absorption spectra of ox-TMB incubated with ICTO and TO in the presence of H₂O₂. **b** Quantitative analysis of absorbance at $\lambda = 652$ nm in the presence of H₂O₂ substrate at different pH conditions (n = 3 independent experiments, data are presented as mean \pm SD). Statistical significance was calculated using one-way analysis of variance (ANOVA) followed by Tukey’s post-hoc test for multiple comparisons. Source data are provided as a Source Data file.

Supplementary Fig. 15. **a** UV-vis absorption spectra of ox-TMB incubated with ICTO and TO in the absence of H₂O₂. **b** Quantitative analysis of absorbance at $\lambda = 652$ nm at different pH conditions ($n = 3$ independent experiments, data are presented as mean \pm SD). Statistical significance was calculated using one-way ANOVA followed by Tukey's post-hoc test for multiple comparisons. Source data are provided as a Source Data file.

Supplementary Fig. 35. CCK-8 assay after series of concentrations of ICTO (**a**) or TO (**b**) interventions to 143b cells or BMSCs ($n = 3$ biologically independent replicates). Data are presented as mean \pm SD, and ns represents no significant difference; statistical significance was calculated using one-way ANOVA followed by Tukey's post-hoc test for multiple comparisons. Source data are provided as a Source Data file.

(8) “*In the osteosarcoma model, how is the size of the 3D-printed scaffold determined and how is the concentration of nanoparticles optimized?*”

Response to this comment:

For the comment of “*In the osteosarcoma model, how is the size of the 3D-printed scaffold determined?*”

We sincerely appreciate your valuable comments and apologize for the lack of a detailed description regarding the design of the scaffold size for *in vivo* experiments. The volume of subcutaneous osteosarcoma-bearing mice was calculated using the formula: (longest diameter) × (shortest diameter)² / 2. Assuming the tumor is approximately spherical (longest diameter = shortest diameter), this formula yields a longest diameter of approximately 7.36 mm. Therefore, the diameter (ϕ) of the implantable scaffold should be < 7.36 mm to allow sufficient space for surgical suturing and manipulation. Based on a review of previous literature and surgical experience, we determined that a scaffold with $\phi = 5$ mm and a thickness of approximately 2 mm is suitable, as it facilitates easy implantation and surgical handling. Additionally, for the *in vivo* bone regeneration experiments, we established bilateral cranial defects with a diameter of approximately 5 mm (*Nat. Protoc.* **7**, 1918-1929 (2012), *Adv. Sci.* **11**, 2400349 (2024)) that meet the definition of critical-sized bone defect, which means the minimum size of bone defect that cannot be spontaneously repaired through natural healing processes in a specific animal model (*J. Craniofac. Surg.* **9**, 310-316 (1998)). Thus, the $\phi = 5$ mm × 2 mm scaffolds ensure consistency between the scaffold parameters used in the osteosarcoma model and those in the cranial bone defect implants.

We hope that these statements and clarifications address your concerns and would like to thank you again for your comments and suggestions.

For the comment, “*how is the concentration of nanoparticles optimized?*”

Thank you for the constructive comments. Regarding the optimized loading amount of ICTO, we have loaded different amount of ICTO nanoparticles on the HA scaffold. Based on the results of *in vitro* antitumor experiments, we demonstrate that HS-ICTO-2.0 is the optimized loading amount for the *in vivo* studies, which shows excellent biocompatibility and significant tumor-killing effects of *in vitro* experiments. The results confirmed that HS-ICTO-2.0, compared with HS-ICTO-0.5 and HS-ICTO-1.0, achieved the most effective tumor suppression when combined with ultrasound while exhibiting no significant cytotoxicity against BMSCs. Therefore, we chose HS-ICTO-2.0 for the

osteosarcoma model. In order to fully stress the biosafety of HS-ICTO-2.0, we evaluate the potential cumulative metal ion toxicity of internalized ICTO nanoparticles with release kinetics analyses *in vitro*. Given the two distinct pathophysiological models of our research, osteosarcoma and cranial defects, which necessitate different therapeutic durations, we employed PBS at pH 6.5 and pH 7.4 to simulate the tumor microenvironment and bone defect condition, respectively. Specifically, HS-ICTO-2.0 was immersed in PBS at both pH levels for four weeks. Supernatants were collected at predetermined time points, and the release concentrations of Ir^{3+} , Ti^{4+} , and Ca^{2+} ions were quantitatively analyzed using inductively coupled plasma atomic emission spectroscopy (ICP-AES). The experimental results revealed exceptional ionic stability, with cumulative leaching rates not exceeding 0.035% over a four-week period under acidic conditions (pH 6.5). Similarly, when subjected to physiological pH conditions (7.4), the cumulative ion dissolution remained below 0.031% throughout the four-week duration, indicating negligible potential for ion-mediated cytotoxicity (Supplementary Fig. 47). Together with the hematological, biochemical, and major organ H&E staining results, we believe that HS-ICTO-2.0 exhibits favorable biocompatibility and demonstrates significant tumor-inhibitory effects after synergistic ultrasound irradiation.

The corresponding data have been added in the revised supplementary information, and relevant results have been discussed as follows:

Page 14 in the revised manuscript, “The effects of ICTO loading amounts on HS scaffolds (HS-ICTO-0.5, 1.0, and 2.0) have also been studied, which presents negligible ion-mediated cytotoxicity and no impairment against the BMSCs’ viability (Supplementary Figs. 46, 47) while showing a dose-dependent manner on inhibiting 143b osteosarcoma cells.”

Page 31 in the revised supplementary information, “Notably, to evaluate the potential cumulative metal ion toxicity of both HS-ICTO and internalized ICTO nanoparticles, we immersed HS-ICTO in PBS. Supernatants were collected at predetermined time points, and the release concentrations of Ir^{3+} , Ti^{4+} , and Ca^{2+} were quantitatively analyzed using inductively coupled plasma atomic emission spectroscopy (ICP-AES). The experimental results reveal exceptional ionic stability, with cumulative leaching rates not exceeding 0.035% over a four-week period under both mildly acidic (pH 6.5) and physiological (pH 7.4) conditions, indicating negligible ionic toxicity potential and substantiating the superior biocompatibility and safety profile of scaffolds.”

(9) “After tumor treatment, can the scaffold degrade naturally, or does it require surgical removal?”

Response to this comment:

We sincerely appreciate your valuable comments. The hydroxyapatite (HA)-based scaffolds used in this study do not require surgical removal and will be gradually replaced by newly formed bone tissue and bone metabolism. After implantation, newly formed bone directly bonds to HA through a carbonated calcium-deficient apatite layer at the bone-implant interface. Its surface supports the adhesion, growth, and differentiation of osteoblasts, and new bone is deposited through the creeping substitution of adjacent living bone. Additionally, its surface can serve as a nucleating site for bone mineralization (*Biomater. Res.* **23**, 4 (2019)). HA would be gradually replaced by newly formed bone tissue through the osteogenic-osteoclastic metabolism after implantation, therefore eliminating the need for surgical removal (*J. Biomed. Mater. Res. A.* **67**, 713-718 (2003)). In the current field of bone tissue engineering, the HA scaffold is a representative bone implant that exhibits excellent properties, including biodegradability, biocompatibility, osteoconductivity, and osseointegration, and has been widely used as an artificial bone substitute. In this study, the coating of ICTO on HA-based scaffold will not influence the aforementioned intrinsic properties of the HA substrates, we demonstrated that ICTO will enhance the osteogenic repair capabilities of HA further. The ICTO coating did not interfere with the adhesion of BMSCs (Fig. 5e) and effectively induced the ingrowth of new bone tissue (Figs. 7e-g). Therefore, the HS-ICTO scaffold can efficiently promote the bone repair process and maintain the gradually degradability of HA scaffold, thus it doesn't require surgical removal.

Fig. 5 e Representative CLSM images of phalloidin-Rhodamine stained BMSCs with different treatments and the corresponding cell size evaluation ($n = 20$, the size of twenty random cells from each group was calculated by ImageJ software). Data are presented as mean \pm SD, and ns represents no significant difference; statistical significance was calculated using one-way ANOVA followed by Tukey's post-hoc test for multiple comparisons.

Fig. 7 e H&E staining and **f** Masson's trichrome staining of regenerated bones induced by different scaffolds. **g** Coronal views of the defect areas from micro-CT images.

(10) “Hydroxyapatite (HA) exhibits certain piezoelectric properties. In the osteosarcoma model, the

scaffold alone has been observed to inhibit tumor growth. Could this effect be potentially related to its piezoelectric characteristics?”

Response to this comment:

We sincerely appreciate your valuable comments and suggestions. Indeed, in this study, we did not consider the potential tumor-killing efficacy resulting from the piezoelectric properties of HA. We fully acknowledge the importance of rigorously verifying the piezoelectric potential in this context. To address this concern, we conducted systematic piezoelectric investigations at both macroscopic and nanoscale levels. Specifically, we initially employed an oscilloscope (RIGOL DS1102 Z-E, RIGOL Technologies, China) to evaluate the macroscopic piezoelectric performance of HS and HA powders. Utilizing the direct piezoelectric effect, periodic mechanical forces were applied while simultaneously monitoring the voltage signals generated by internal charge separation. Experimental results indicated that the HS and HA powders used in this study did not generate significant voltage signals under mechanical stimulation, preliminarily suggesting the lack of notable piezoelectric effects of the used HS and HA powders for this study (Supplementary Fig. 58a).

Furthermore, to validate piezoelectric characteristics at nanoscale, quantitative characterization was performed using an atomic force microscope (AFM, MFP-3D-BIO, Asylum Research, USA) on HS and HS-ICTO samples. The specimens were fabricated into standardized circular discs with a diameter of 5 mm. Three non-edge regions were randomly selected for each sample group and subjected to alternating electric fields to detect surface microdeformations, thereby verifying nanoscale piezoelectric responses. Each location was measured in triplicate to mitigate local heterogeneity effects. As illustrated in Supplementary Fig. 58b, c, no characteristic waveforms or images indicative of piezoelectric properties were observed in either HS or HS-ICTO, further corroborating the absence of significant piezoelectric activity in HA-based materials under our experimental conditions. Therefore, we suppose the tumor inhibition effects in HS and HS + US groups may be attributed to the subtle chemical effects of nano-HA, such as calcium ion overload (*Nano Research*. **15**, 6256-6265 (2022)).

Page 17 in the revised manuscript: “Considering the potential piezoelectric attributes of HA, we employed an oscilloscope and atomic force microscopy to investigate its piezoelectric properties (Supplementary Fig. 58). The results indicate a negligible presence of piezoelectricity in both HS and HS-ICTO. Thus, we hypothesize that the minor tumoricidal effects observed in HS and HS + US groups may be attributed to the subtle chemical effects of nano-HA, potentially involving calcium ion

overload^{5,77}.”

Macroscopic piezoelectric response detected via an oscilloscope

Nanoscale piezoelectric property detected via AFM

Supplementary Fig. 58. a Oscilloscope detection of electrical signals generated under vibration. Representative atomic force detection images and longitudinal signal piezoresponse phase (LS PR Phase) waves of HS (b) and HS-ICTO (c). Source data are provided as a Source Data file.

(11) “The inclusion of staining images for Caspase-3 in vivo is recommended, as they are crucial for confirming the apoptosis process in vivo.”

Response to this comment:

We are grateful for your feedback, which undoubtedly improves the quality of our work. Based on your suggestion, we have prepared new tumor tissue sections and performed immunofluorescence staining for cleaved caspase-3 (CST, Rabbit mAb #9664) to label apoptotic cells specifically. The results of the positive cell rate analysis revealed no significant differences in cleaved caspase-3-positive cells among the Sham, HS, and HS + US groups. In contrast, the HS-ICTO group exhibited a marked increase in the number of positive cells, and the HS-ICTO + US group showed nearly complete occupancy of the sections by positive cells. Combined with the TUNEL staining results, these findings robustly demonstrate that the HS-ICTO + US treatment effectively induces tumor cell apoptosis. The corresponding data have been incorporated into Supplementary Fig. 63 and relevant results have been discussed as follows:

Page 18 in the revised manuscript: “To further validate the apoptotic status of cells within the tumor tissue, we employed cleaved caspase-3 immunofluorescence staining which specifically labels apoptotic cells (Supplementary Fig. 63). The results of the positive cell rate analysis reveal no significant differences in cleaved caspase-3-positive cells among the Sham, HS, and HS + US groups. In contrast, the HS-ICTO group exhibits a marked increase in the number of positive cells, and the HS-ICTO + US group shows nearly complete occupancy by positive cells. Combined with the TUNEL staining results, these findings robustly demonstrate that the HS-ICTO + US treatment effectively induces tumor cell apoptosis.”

Supplementary Fig. 63. **a** Representative cleaved caspase-3 fluorescence staining images of the tumors from different groups. The images were representative of five independently repeated

experiments from each group. **b** Quantitative analysis of the cleaved caspase-3-positive cell ratios (n = 5 biologically independent replicates). Data are presented as mean ± SD, and ns represents no significant difference; statistical significance was calculated using one-way ANOVA followed by Tukey's post-hoc test for multiple comparisons. Source data are provided as a Source Data file.

(12) “On page 26, it is essential to include appropriate references to support the discussion on differential gene expression following HS-ICTO treatment.”

Response to this comment:

We sincerely appreciate your valuable suggestion and would like to apologize for the lack of a detailed presentation regarding the discussion on differential gene expression following HS-ICTO treatment. The differentially expressed genes mentioned in the manuscript were selected as representative genes from the GO terms enriched through the DAVID database. We have now incorporated relevant references (Ref. 86 to Ref. 93) to functional studies related to these genes, thereby enhancing the context and substantiation of our findings.

Page 26 in revised manuscript: “It has been found that HS-ICTO treatment significantly suppresses the gene expression that related to inflammatory responses, including *CXCL6* and *CCL21* for neutrophil chemotaxis⁸⁶; *TNF* and *IL1B* for positive regulation of I-kappaB kinase/NF-kappaB signaling⁸⁷; bone resorption (*NFATC1* and *OSCAR* for osteoclast differentiation⁸⁸; *MMP14* and *MMP9* for collagen catabolic process⁸⁹); and ROS-related responses (*PRKAA1* and *GPXI* for response to hydrogen peroxide^{90,91}; *ABCCI* and *BTGI* for response to oxidative stress^{92,93}).”

Reviewer #2:

General comment: “This manuscript presents an interesting 3D-printed scaffold with sono-activable and biocatalytic properties for intelligently sequential therapies in osteosarcoma eradication and defect regeneration after surgical implantation. Such therapeutic systems display superior, spatiotemporally controllable, and versatile H₂O₂-catalytic performances, which promptly generates

ROS in TME via multienzyme-like pathways in conjunction with sono-activation. Concurrently, it can intelligently switch to catalyze H₂O₂ to O₂ for effectively blocking endogenous H₂O₂-mediated oxidative stress during the subsequent bone defect regeneration. This research provides a new and promising strategy for the engineering of multifunctional bone implants with simultaneous preclusion of tumor recurrence and promotion of bone repair based on ROS-catalytic regulation, offering essential guidance for the realization of precise oncological intervention and tissue regeneration. The results are innovative and the manuscript is well-structured, containing extensive and comprehensive studies and convincing experimental evidences. Overall, I would like to recommend the publication of the work in Nature Communications after addressing the following issues:”

Response to the general comment:

Thank you for your valuable comments and suggestions, which have greatly enhanced the quality of this research article. The manuscript has been significantly revised in accordance with the reviewers' comments. All necessary experimental conditions and required data have been included, and all the questions and considerations have been well-addressed in the revised manuscript and supplementary information.

(1) *“For the rationale for constructing biocatalysts with multi-edge petaloid microstructure, more explanation is needed. For example, what specific advantages does this structural characteristic confer to their performance or application?”*

Response to this comment:

We sincerely appreciate the valuable comment. The fabricated ICTO nanoparticles with multi-edge petaloid microstructures demonstrate distinctive structural advantages, encompassing multi-dimensional hierarchical porous architecture, accessible open surfaces, nanoscale petaloid sheets, and optimized specific surface area. Their significant promotion of H₂O₂-based biocatalytic processes and applications manifests in three primary aspects. 1) Enhanced catalytic efficiency; the open channels and accessible surface topology facilitate superior H₂O₂ substrate diffusion kinetics, reducing spatial constraints and ensuring efficient catalysis. Additionally, the high specific surface area and multi-dimensional hierarchical structure maximize Ir catalytic sites availability, enhancing reaction velocity and overall ROS generation efficiency (*Adv. Funct. Mater.* **32**, 2111171 (2022), *ACS Nano* **17**, 6731–6744 (2023)). 2) Efficient energy absorption; the petaloid architecture, attributable to its enhanced

surface area and hierarchical organization, effectively absorbs external ultrasonic energy, driving efficient electron transfer processes and subsequently promoting sono-activatable ROS amplification (*Matter* **6**, 2206-2234 (2023)). 3) Improved adhesion and stability; the multi-edged structure significantly increases surface roughness, strengthening mechanical adhesion to the 3D-printed HS substrate, thereby ensuring stable and sustained therapeutic efficacy. Fig. 2a in the revised manuscript systematically elucidates the structural characteristics and functional advantages of ICTO nanoparticles, with supplementary discussion provided as follows:

Page 9 in the revised manuscript: “Leveraging the distinctive 3D hierarchical petaloid nanosheet microstructure with optimized specific surface area, ICTO guarantees superior H₂O₂ substrate accessibility, optimal exposure of surface Ir active sites and enhanced ultrasonic energy absorption, which may result in remarkable reactivities in sono-activatable ROS biocatalytic processes (Fig. 2a).”

Fig. 2 a A comprehensive schematic representation of the advantageous structural characteristics of ICTO and the resultant versatile enzyme-like biocatalytic activities.

(2) “In Fig. 1j, why was Ir/C selected as a reference in the Ir 4f analysis, and what insights does it provide into the electronic structure of the Ir site in ICTO? This is not clearly explained.”

Response to this comment:

Thank you for the constructive comments. We will elucidate the electronic structure

characteristics attributes of two different substrates to justify the selection of Ir/C as a reference. The carbon-based matrix in Ir/C, while providing exceptional surface area and robust physical support for Ir cluster dispersion, maintains a relatively stable electronic configuration with simple surface chemical environments. The anchoring mechanism of Ir clusters primarily occurs through π - π interactions, which inherently possess limited electron-donating capabilities, thus resulting in minimal disturbance of the electron density at Ir active sites and preserving the intrinsic electronic structural properties of the Ir sites (*Nat. Catal.* **2**, 955–970 (2019), *Science* **386**, 915–920 (2024)). In contrast, titanium dioxide (TO) in ICTO demonstrates distinctive redox properties and polarity, establishing strong Ir-O-Ti chemical coupling with Ir clusters to facilitate metal-support electronic interactions, driving robust interfacial charge transfer. This interaction not only enhances the stability of Ir sites but also promotes the formation of electron-rich structures through electron redistribution mechanisms. We posit that the significant difference in electronic behavior between these substrates provides a compelling rationale for employing Ir/C as a reference standard for Ir 4f XPS analysis, thereby enabling the elucidation of the TO substrate's influence on the electronic structure and characteristics of supported Ir clusters.

(3) “For the mechanism of intelligent performance conversion of ICTO, it is necessary to further clarify the relationship between it and biological applications.”

Response to this comment:

We appreciate your valuable and constructive feedback on enhancing the manuscript quality, and we acknowledge the previous inadequate explanation. We concur that investigating the underlying reasons governing different physiological environment-induced intelligent performance conversion is crucial for elucidating catalytic mechanisms and their biological applications. Further details will be provided in the subsequent discussion.

During osteosarcoma progression, tumor cells undergo rapid proliferation within the tumor tissue, utilizing anaerobic glycolysis as their primary energy source, consequently generating an acidic tumor microenvironment (TME) characterized by H₂O₂ overexpression (*Natl. Sci. Rev.* **5**, 269–286 (2018), *Coord. Chem. Rev.* **481**, 215049 (2023)). In the context of bone defect regeneration, elevated H₂O₂ levels persist due to inflammatory mediators and immune cell infiltration while maintaining neutral or slightly alkaline pH during cellular proliferation and tissue remodeling phases (*Nature* **459**, 996–999

(2009), *J. Cell. Biochem.* **115**, 427–435 (2014).). Therefore, the proposed intelligent performance switching of ICTO in our study is triggered based on pH microenvironment variations.

Density functional theory (DFT) calculations reveal the critical role of hydrogen ion (H^+) in ROS generation. As shown in Fig. 2h and Supplementary Figs. 32, 33, in stage 3 (S3) of the POD-like pathway, $*OH$ at the Ti site captures an H^+ proton to form H_2O and depart from the surface, resetting the catalyst for the next cycle. Comparatively, in the absence of H^+ during the reaction, $*OH$ interacts with another H_2O_2 to generate $*H_2O$ and $*OOH$ intermediates. The subsequent desorption of $*OOH$ encounters a significant thermodynamic energy barrier of 1.81 eV, emphasizing the crucial role of an acidic environment in promoting the regeneration and recycling of catalytic Ir sites, as well as being essential for optimal ROS production. Under neutral conditions, H_2O_2 , as the sole proton source, preferentially follows the CAT-like oxygen generation pathway, enabling performance-switching functionality. Moreover, according to our experimental studies, ICTO enables pH-dependent ROS generation activity, predominantly producing ROS under mildly acidic conditions (similar to the pH values in TME) and negligible ROS effect observed in neutral or alkaline conditions. Furthermore, we also validated this at the cellular level. Specifically, ICTO generates substantial ROS in 143b osteosarcoma cells, ultimately inducing oxidative stress-mediated cytotoxicity. However, ICTO is unable to generate significant ROS in rat bone marrow mesenchymal stem cells (BMSCs, pH 7.4), showing excellent viability (96.5% at $150 \mu g mL^{-1}$), thus implying the relevance of ROS generation and cytotoxicity to the specific TME conditions (Supplementary Fig. 35).

In conclusion, ICTO demonstrates distinct excellent catalytic properties based on pathological microenvironment differences to achieve intelligent switching on ROS generation or ROS scavenging, thus simultaneously satisfying the requirements for ROS-mediated antitumor activity and endogenous H_2O_2 elimination with oxygen generation for anti-inflammatory and osteogenic responses. The corresponding data have been added to the revised supplementary information, and relevant statements have been supplemented in the revised manuscripts, which are also shown as follows:

Page 12 in the revised manuscript: “Followingly, H_2O desorbs from the Ir site, retaining the initial $*OH$ on the Ti site, which in turn captures an H^+ proton from the acidic environment to form H_2O and depart from the surface, resetting the catalyst for the next cycle (Figs. 2h, i). Comparatively, in the absence of H^+ during the reaction, $*OH$ interacts with another H_2O_2 to generate $*H_2O$ and $*OOH$ intermediates. The subsequent desorption of $*OOH$ encounters a significant thermodynamic energy

barrier of 1.81 eV, emphasizing the necessity of an acidic environment in promoting the biocatalytic activity for optimal ROS production, which aligns well with prior experimental outcomes (Supplementary Figs. 32, 33).”

Supplementary Fig. 32. Proposed reaction POD-like and CAT-like pathways on ICTO. Path 2 in POD-like mechanisms: H⁺ protons involvement; Path 2 in POD-like mechanisms: reaction with another H₂O₂ to form *H₂O and *OOH intermediates.

Supplementary Fig. 33. Calculated the free energy for the liberation of •OOH in step iv by ICTO in a POD-like pathway. Source data are provided as a Source Data file.

Supplementary Fig. 35. CCK-8 assay after series of concentrations of ICTO interventions to 143b cells or BMSCs (n = 3 biologically independent replicates). Data are presented as mean \pm SD, and ns represents no significant difference; statistical significance was calculated using one-way ANOVA followed by Tukey’s post-hoc test for multiple comparisons. Source data are provided as a Source Data file.

(4) “Fig. 5f and supplementary Fig. 45 illustrate the endocytosis of ICTO nanoparticles. The authors may discuss whether this process poses a risk of cumulative metal ion toxicity.”

Response to this comment:

Thank you for your thoughtful review and insightful suggestions. To evaluate the potential cumulative metal ion toxicity of internalized ICTO nanoparticles, we initially conducted release kinetics analyses *in vitro*. Given the two distinct pathophysiological models of our research, osteosarcoma and cranial defects, which necessitate different therapeutic durations, we employed PBS at pH 6.5 and pH 7.4 to simulate the tumor microenvironment and bone defect condition, respectively. Specifically, HS-ICTO was immersed in PBS at both pH levels for four weeks. Supernatants were collected at predetermined time points, and the release concentrations of Ir^{3+} , Ti^{4+} , and Ca^{2+} ions were quantitatively analyzed using inductively coupled plasma atomic emission spectroscopy (ICP-AES). The experimental results revealed exceptional ionic stability, with cumulative leaching rates not exceeding 0.035% over a four-week period under acidic conditions (pH 6.5). Similarly, when subjected to physiological pH conditions (7.4), the cumulative ion dissolution remained below 0.031%

throughout the four-week duration, indicating negligible potential for ion-mediated cytotoxicity (Supplementary Fig. 47).

To further verify the potential toxicity of metal ions that may exist in the *in vivo* experiments, we conducted subcutaneous implantation of HS and HS-ICTO in the dorsal region of 8-week-old male Sprague-Dawley (SD) rats. Peripheral blood was collected after 4 weeks for comprehensive hematological and biochemical analyses of liver and kidney functions (Supplementary Fig. 61). The results demonstrated that key hematological parameters (RBC, HGB, PLT, and WBC) in both groups remained within the normal physiological range, with no significant inflammatory or anemic alterations observed. Furthermore, biochemical assays revealed no notable abnormalities in critical indicators such as ALT, AST, UREA, and CREA, thereby substantiating the circulatory safety of HS and HS-ICTO over extended metabolic periods. The relevant findings have been discussed in the revised manuscript as follows:

The corresponding data have been added in Supplementary Figs. 47, 61 in the revised supplementary information and relevant results have been discussed as follows:

Page 14 in the revised manuscript, “The effects of ICTO loading amounts on HS scaffolds (HS-ICTO-0.5, 1.0, and 2.0) have also been studied, which presents negligible ion-mediated cytotoxicity and no impairment against the BMSCs’ viability (Supplementary Figs. 46, 47) while showing a dose-dependent manner on inhibiting 143b osteosarcoma cells.”

Page 17 in the revised manuscript: “Moreover, to further underscore the biosafety profile of HS-ICTO, we conducted subcutaneous implantation of HS and HS-ICTO in the dorsal region of 8-week-old male Sprague-Dawley (SD) rats. Peripheral blood was collected after 4 weeks for comprehensive hematological and biochemical evaluations of liver and kidney functions (Supplementary Fig. 61). The results demonstrate that parameters including RBC, HGB, PLT, and WBC in both groups remain within the normal physiological range, with no significant inflammatory or anemic alterations observed. Furthermore, liver/kidney function biochemical assays reveal no notable abnormalities in critical indicators such as ALT, AST, UREA, and CREA, thereby substantiating the circulatory safety of HS and HS-ICTO over extended metabolic periods.”

Page 30 in the revised supplementary information, “Notably, to evaluate the potential cumulative metal ion toxicity of both HS-ICTO and internalized ICTO nanoparticles, we immersed HS-ICTO in PBS. Supernatants were collected at predetermined time points, and the release concentrations of Ir^{3+} ,

Ti⁴⁺, and Ca²⁺ were quantitatively analyzed using inductively coupled plasma atomic emission spectroscopy (ICP-AES). The experimental results reveal exceptional ionic stability, with cumulative leaching rates not exceeding 0.035% over a four-week period under both mildly acidic (pH 6.5) and physiological (pH 7.4) conditions, indicating negligible ionic toxicity potential and substantiating the superior biocompatibility and safety profile of scaffolds.”

Supplementary Fig. 47. Percentage of metal release at different immersion times based on HS-ICTO-2.0 scaffolds. Source data are provided as a Source Data file.

Supplementary Fig. 61. a Blood routine test results including red blood cell count (RBC), hemoglobin (HGB), platelets (PLT), and white blood cell count (WBC). **b** Biochemistry test results including alanine aminotransferase (ALT), aspartate aminotransferase (AST), urea, and creatinine (CREA) (n = 3 biologically independent replicates). Data are presented as mean \pm SD, and ns represents no significant difference; statistical significance was calculated using two-tailed Student's *t*-test. Source data are provided as a Source Data file.

(5) “Please provide more comments or experiments to clarify whether the mechanical strength of the prepared 3D scaffold is sufficient to fill large segmental bone defects.”

Response to this comment:

Thank you for your valuable comments. The mechanical properties of the scaffolds were evaluated using a universal mechanical testing system. Cylindrical specimens (6 mm in diameter, 9 mm in height) were compressed at a constant displacement rate of 1 mm/min until visible fracture or mechanical failure (defined as a significant load drop) occurred. The results revealed no statistically significant difference in compressive strength between HS (2.01 ± 0.29 MPa) and HS-ICTO (2.02 ± 0.15 MPa). This indicates that the ICTO coating did not compromise the structural integrity or compressive resistance of the HS matrix. Stress-strain curves further corroborated these findings, with both groups exhibiting characteristic brittle fracture behavior and nearly identical deformation profiles. The overlapping trajectories of the curves confirm comparable mechanical responses under compressive loading. According to these data, both HS and HS-ICTO exhibit remarkable bioactivity and osteoconductive properties, with their compressive strength matching the mechanical properties of human cancellous bone (2-12 MPa). In clinical applications, these materials have been successfully employed for large-segment bone defect reconstruction in non-load-bearing anatomical regions, such as craniomaxillofacial defect filling. It should be emphasized that for load-bearing site reconstruction (e.g., femoral or tibial defects), where complex dynamic mechanical loading (including compressive, bending, and torsional stresses) must be sustained, standalone calcium phosphate ceramics application cannot meet biomechanical requirements. Therefore, current clinical protocols generally adopt a composite repair strategy: establishing initial mechanical support through internal fixation systems (e.g., locking compression plates, intramedullary nails) while utilizing calcium phosphate ceramics scaffolds to facilitate bone regeneration via their osteoinductive properties (*Adv. Sci.* **12**, e2408459

(2025)). During the healing process, new bone tissue undergoes “stress-adaptive remodeling”, undergoing progressive mineralization to achieve gradual biomechanical load transfer.

The corresponding data have been added to the revised supplementary information, and relevant statements have been supplemented in the revised manuscripts, which are also shown as follows:

Page 6 in revised manuscript: “To ascertain whether the mechanical properties of the HS-ICTO scaffold meet the standards for bone tissue implants, we employed a universal mechanical testing system. The findings reveal that HS and HS-ICTO demonstrate nearly identical deformation profiles. In particular, the variance in compressive strength between HS (2.01 ± 0.29 MPa) and HS-ICTO (2.02 ± 0.15 MPa) display no statistically significant difference, which suggests that the ICTO coating does not compromise the structural integrity or compressive properties of the HS matrix and is comparable to the mechanical properties of human trabecular bone (Supplementary Fig. 3).”

Supplementary Fig. 3. a Stress-strain curves and **b** Compressive strength test of HS and HS-ICTO ($n = 5$ independent experiments, data are presented as mean \pm SD). Source data are provided as a Source Data file.

(6) “Page 20, the authors proposed, “...the excessive H_2O_2 and insufficient O_2 supply may lead to cascaded side effects on the osteogenesis process of BMSCs by restraining motility...” It is necessary to supplement cell migration-related experiments to evaluate the ability of HS-ICTO to recruit stem cells.”

Response to this comment:

We sincerely appreciate your valuable and insightful comments. We apologize for the lack of

experimental validation regarding the proposed mechanism of H₂O₂-mediated regulation of BMSC motility in our previous manuscript. According to your comment, we have explored the ability of HS-ICTO to protect BMSC migration using the Transwell assay system. Briefly, BMSCs (3 × 10⁴ cells/well) were seeded in the upper chamber of a Transwell insert (8 μm pore size). The lower chamber was filled with a complete medium containing 100 μM H₂O₂, and either HS or HS-ICTO scaffolds were placed in the lower chamber. After 12 hours, the cells in the upper chamber were removed using a cotton swab, and the cells that migrated to the lower surface of the membrane were stained with crystal violet. The crystal violet-positive area was quantified using ImageJ to reflect the extent of cell migration. The results demonstrated that the migrated cells were reduced in the HS + H₂O₂ group, whereas the HS-ICTO + H₂O₂ group showed a significant improvement in the number of migrated cells. This confirms that HS-ICTO can protect the migration ability of BMSCs under H₂O₂ stimulation.

The corresponding data have been added to the revised supplementary information and relevant statements have been supplemented in the revised manuscripts, which are also shown as follows:

Page 21 in the revised manuscript: “Additionally, the motility of BMSCs was analyzed through transwell-based migration assay (Supplementary Fig. 67). The results demonstrate that the migrated cells are reduced in the HS + H₂O₂ group, whereas the HS-ICTO + H₂O₂ group shows a significant improvement in the crystal violet-labelled migrated cells. This confirms that HS-ICTO can protect the migration ability of BMSCs under H₂O₂ stimulation.”

Supplementary Fig. 67. **a** Representative crystal violet-labeled migrated BMSCs images. **b** Illustration of the transwell migration experiment of BMSCs. The BMSCs. were seeded in the upper chamber, and the ROS-conditioned complete medium were added in the lower chamber. HS or HS-ICTO was placed in the lower chamber. **c** Quantitative analysis of the crystal violet area (%) ($n = 3$ biologically independent replicates). Data are presented as mean \pm SD; statistical significance was calculated using one-way ANOVA followed by Tukey's post-hoc test for multiple comparisons. Source data are provided as a Source Data file.

(7) “Osteosarcoma typically occurs in large weight-bearing bones such as the femur, but the authors choose a cranial bone defect model to simulate postoperative bone defect repair instead of a weight-bearing bone defect model. So the authors are suggested to provide the connection of these animal models.”

Response to this comment:

We sincerely appreciate the reviewer's insightful comment regarding the rationale for selecting a cranial bone defect model in our study. We fully agree that osteosarcoma predominantly arises in weight-bearing bones such as the femur, and we acknowledge the importance of evaluating therapeutic

strategies in clinically relevant anatomical contexts. Below, we clarify our rationale for choosing the cranial bone defect model in this work while emphasizing our commitment to addressing this point in future studies.

Firstly, we employed a rat cranial critical-sized defect model, which means the minimum size of the bone defect that cannot be spontaneously repaired through natural healing processes in a specific animal model (*J. Craniofac. Surg.* **9**, 310-316 (1998)). This model exhibits characteristics that are largely consistent with those of large segmental bone defects and can effectively reflect the osteoinductive, osteoconductive, osseointegration, and biocompatibility properties of implanted materials. As mentioned earlier, the CaP material used in this study has demonstrated its potential as a filler for large segmental bone defects and has already been utilized in clinical applications with support from internal fixation systems (*Adv. Sci.* **12**, e2408459 (2025)). Therefore, in terms of efficacy evaluation, the repair of rat cranial critical-sized defects can, to some extent, reflect the therapeutic effectiveness of the HS-ICTO scaffold in addressing difficult-to-heal bone defects.

Additionally, the cranial bone defect model was selected for its well-established utility in the controlled evaluation of bone regeneration and repair mechanisms. Unlike weight-bearing bones, cranial defects minimize confounding variables such as mechanical stress and dynamic loading, allowing us to isolate and characterize the biological efficacy of our therapeutic approach (e.g., ROS scavenging, anti-inflammation, osteogenic activity, and biomaterial integration). This model is widely adopted in foundational bone regeneration studies (*Nat. Protoc.* **7**, 1918-1929 (2012)) due to its reproducibility, surgical accessibility, and reduced risk of postoperative complications, which were critical for our proof-of-concept investigation.

We fully agree that validating our findings in a weight-bearing bone model (e.g., femur or tibia) is essential for clinical translation. In future studies, we aim to optimize the application scope of HS-ICTO by employing it to fill segmental bone defect models constructed in the weight-bearing bones of large animals (e.g., the femur of beagle dogs), thereby evaluating its reparative effects in weight-bearing bones.

We thank all referees again for their helpful comments and suggestions, and hope that this significantly revised manuscript is now acceptable for publication in *Nature Communications*.

Best Regards,

Yours Sincerely,

Prof. Dr. Chong Cheng (on behalf of the authors)